# Study on Characteristics and Control of Aerodynamic Noise of a High-Speed Centrifugal Air Compressor for Vehicle Fuel Cells

**Shizhong Sun** [1],*, **Ziwen Xing** [1],*, **Wenqing Chen** [2], **Minglong Zhou** [2], **Chuang Wang** [1] and **Hanyang Cui** [1]

1  School of Energy and Power Engineering, Xi'an Jiaotong University, Xi'an 710049, China
2  Suzhou Academy, Xi'an Jiaotong University, Suzhou 215123, China
*  Correspondence: sdzcssz@163.com (S.S.); zwxing@mail.xjtu.edu.cn (Z.X.); Tel.: +86-183-9288-8128 (S.S.)

**Abstract:** As the main noise source in the hydrogen fuel cell system, the noise level of the centrifugal air compressor greatly affects the comfort of the hydrogen fuel cell vehicle. For reducing the noise level of centrifugal air compressors, the noise characteristics and control of a high-speed two-stage compressor prototype are studied in this paper. Firstly, the near-field noise measurement, along with the independent component analysis, is carried out to identify the noise source of the developed compressor. Results showed that the "buzz-saw" noise at the rotating fundamental frequency and its low order harmonic frequency in the aerodynamic noise is prominent in the noise spectrum. Thus, the aerodynamic noise characteristics are predicted and analyzed using the CFD–BEM coupling aeroacoustic calculation model. Based on the analysis results, a noise control method coupling the structure optimization and perforated muffler is proposed. The results show that the sound pressure level of the air compressor at 1 m away from the surface is reduced by 4.1 dBA after the structural optimization. A perforated muffler applied in the pipe system of the air compressor can accomplish a reduction of 5.8 dBA in the sound pressure level of the air compressor by impeding the noise transmission on the path. With the coupled noise control methods above, the sound pressure level of the air compressor is reduced from 78.8 dBA to 68.9 dBA under the rated condition.

**Keywords:** fuel cell; two-stage centrifugal air compressor; aerodynamic noise; CFD–BEM coupling analysis; independent component analysis; perforated muffler



## 1. Introduction

The fuel cell has become one of the main ways to solve the current energy crisis, and hydrogen fuel cell vehicles have also become the focus of research in clean-energy vehicles. As the main noise source in the hydrogen fuel cell system, the noise level of the centrifugal air compressor greatly affects the comfort of the hydrogen fuel cell vehicle. The noise of centrifugal air compressors includes aerodynamic noise, electromagnetic noise, and mechanical noise.

Scholars have carried out much research on the aerodynamic noise of the centrifugal air compressor. Raitor et al. [1] decomposed the aerodynamic noise into discrete single-tone noise (blade passing frequency and its harmonics), tip clearance noise, "buzz-saw" noise (shaft rotation frequency and its harmonics), and broadband noise. Among these, the discrete single-tone noise is represented by the noise peak at the blade passing frequency and its harmonics, which is more obvious under all operating conditions. Subsequent scholars researched the discrete single-tone noise of centrifugal compressors [2–9]. The adopted aeroacoustic numerical simulation methods, including the acoustic analogy method [2,3,7,8], broadband noise method [4], CFD (computational fluid dynamics)–BEM (boundary element method) coupling method [5,6,9], etc. McAlpine et al. [10] studied the characteristics and the generation mechanism of the "buzz-saw" noise of a turbofan engine and considered that the "buzz-saw" noise originates from the epitaxial shock wave propagating in the direction of flow generated by the leading edge of the blade under the

condition of tip supersonic speed. Among the noise characteristics of the research object, the discrete single-tone noise is more prominent.

The fuel cell high-speed centrifugal air compressor for vehicles has the characteristics of miniaturization, high speed, and many blades. It is commonly used in the form of two-stage applications. Because its operating conditions and structural parameters are significantly different from those of the centrifugal air compressor used for turbochargers, which are shown in Table 1, its aerodynamic noise characteristics and the improvement of the air compressor aimed at aerodynamic noise are worth investigating. There are few studies on the noise of the centrifugal air compressor for fuel cells. Zuo Shuguang et al. [11] studied the noise characteristics of a centrifugal fan for fuel cells and determined that the main noise components are the suction noise of the centrifugal fan and the aerodynamic noise of the impellers and cooling fan. Wei Kaijun et al. [12] tested and analyzed the noise characteristics of the centrifugal air compressor for fuel cell vehicles, and then analyzed the generation mechanism of a narrow-band howling noise under low-flow conditions with the CFD method. These studies mainly explore the mechanism of noise generation and, as such, lack air compressor improvement for the aerodynamic noise. In this paper, the noise characteristics of a high-speed two-stage centrifugal air compressor for vehicle fuel cells are tested, the independent component analysis method is used to identify the sound source, and the CFD–BEM coupling model is used to calculate the aerodynamic noise. The structure of the compressor is optimized for the improvement of the noise based on the calculation results. The improvement effect is further verified by the acoustic calculation and the test results. In order to reduce the amplitude of the pressure pulsation and impede the noise transmission, a perforated muffler is applied in the inter-stage pipeline of the air compressor. Finally, the whole chain of key technologies from the sound source identification, and the induction mechanism to the noise control, are realized. It provides acoustic theoretical support and a practical basis for the forward development and upgrading of vehicle fuel cell air compressors.

**Table 1.** Operating conditions and structural parameters of the compressors in references.

| References | Number of Blades | Designed Rotation Speed [rev·min⁻¹] | Designed Pressure Ratio | Designed Mass Flow Rate [kg·s⁻¹] |
|---|---|---|---|---|
| [1] | Main: 13; Splitter: 13 | 50,000 | 4.0 | 2.8 |
| [2] | 8 | 4800 | | |
| [3] | 20 | 14,000 | | 5.32 |
| [4] | 32 | 2900 | 1.02 | 1.25 |
| [5,6] | Main: 8; Splitter: 8 | 22,000 | 3.85 | |
| [9] | Main: 7; Splitter: 7 | 98,529 | | 0.211 |

## 2. Noise Characteristics and Source Identification

The research object of this paper is a two-stage centrifugal air compressor used in the air supply system for vehicle fuel cells. Figure 1 shows the schematic diagram of a typical hydrogen fuel cell system for vehicles. In order to increase the efficiency of the fuel cell, the air is compressed by a centrifugal compressor and then enters the fuel cell. Pressurized air reacts with hydrogen in the fuel cell to generate electric energy, which is supplied to the motor or battery.

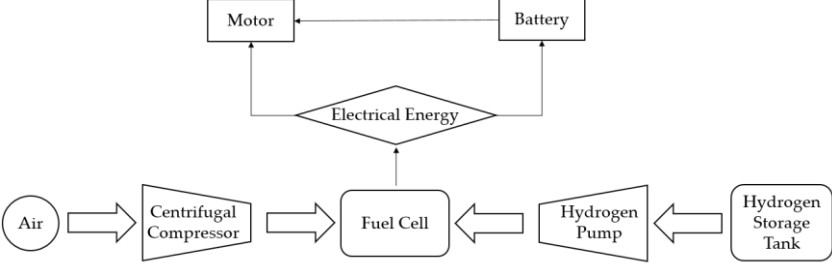

**Figure 1.** Schematic diagram of a typical hydrogen fuel cell system for vehicles.

The cross-section of the centrifugal compressor prototype is shown in Figure 2. The air is sucked and accelerated by the first stage impeller. After being pressurized by the first stage vaneless diffuser and the first stage volute, it enters the inter-stage pipeline and is further accelerated by the second stage impeller. The pressurization and discharge are completed by the second stage vaneless diffuser and the second stage volute. The impellers of both stages have no splitter. The main structural parameters of the air compressor are shown in Table 2. The suction pressure is 101.3 kPa. The specific operating conditions are shown in Table 3. The three conditions are all near the design condition with different rotation speeds.

**Table 2.** Main parameters of the compressor.

| Main Parameters | First Stage | Second Stage |
|---|---|---|
| Diameter of impeller inlet [mm] | 19.3 | 17.5 |
| Diameter of impeller outlet [mm] | 66.2 | 67 |
| Width of impeller outlet [mm] | 4 | 3 |
| Number of the blades | 15 | 15 |

**Table 3.** Operating conditions.

| | Idle Condition | Common Condition | Rated Condition |
|---|---|---|---|
| Rotation speed [r·min$^{-1}$] | 40,000 | 60,000 | 90,000 |
| Discharge pressure [kPa] | 110 | 180 | 250 |
| Mass flow rate [g·s$^{-1}$] | 40 | 95 | 125 |

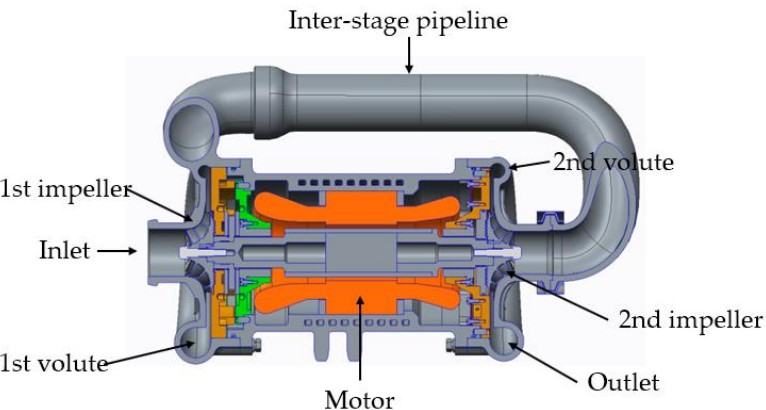

**Figure 2.** Cross-section of the compressor.

### 2.1. Test of Noise Characteristics

The thermal performance and noise characteristics of the air compressor are tested. The flow, pressure ratio, and power consumption of the compressor are tested by the performance test rig of the vehicle fuel cell air compressor. The performance test rig is installed in a standard semi-anechoic chamber. While measuring the performance, the near-surface noise characteristics at 20 mm from eight measuring points on the surface of the compressor are measured. The schematic diagram of the test rig and the positions of the near-field noise measuring points are shown in Figure 3. The noise acquisition equipment is a NL-52 sound level meter produced by Nippon RIKEN from Tokyo, Japan, with a measurement frequency range of 20–20,000 Hz and a resolution of 2.5 Hz. The real images of the test rig and the sound level meter are shown in Figure 4.

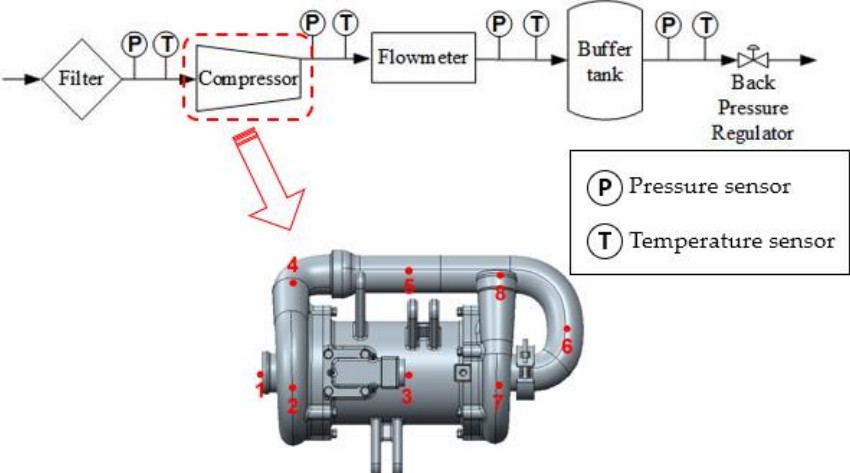

**Figure 3.** Schematic diagram of the test rig and the positions of the noise measuring points.

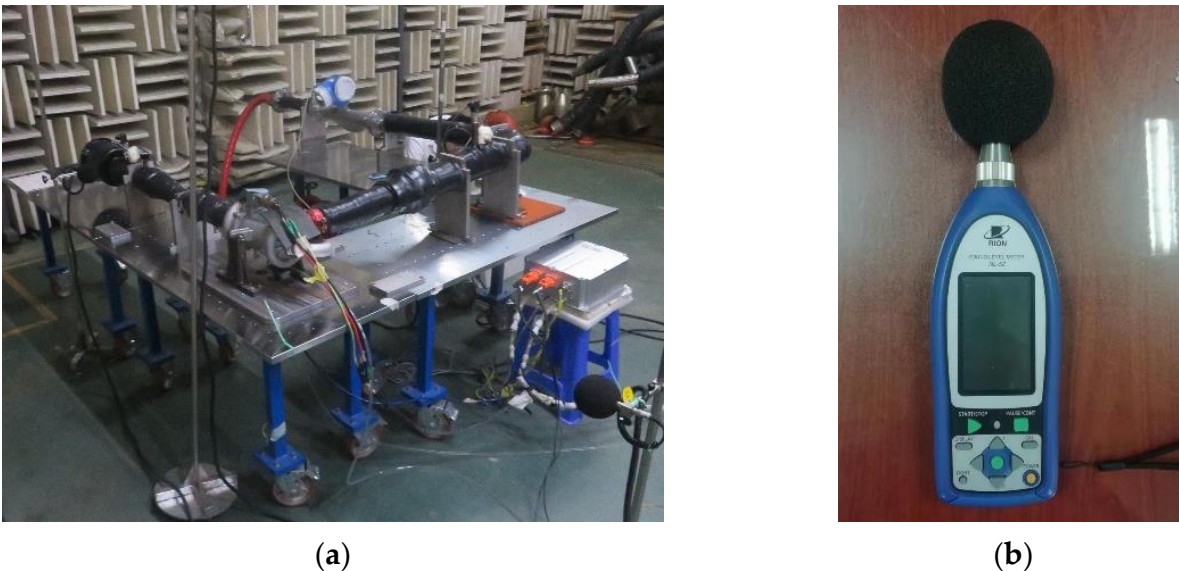

| (**a**) | (**b**) |

**Figure 4.** Real images of the test rig and the sound level meter (**a**) Test rig. (**b**) Sound level meter.

The near-surface noise measurement can alleviate the distortion of an acoustic signal caused by self-attenuation and external interference in the transmission process and, thus, reflect the basic characteristics of the noise radiated from the air compressor as truly as possible. Table 4 shows the noise pressure levels of different measuring points under different working conditions. With the increase in impeller rotation speed, the sound pressure level at each measuring point increases. The sound pressure level of Measuring Point 3 near the motor is relatively high, but it is lower than that of Measuring Points 1, 2, and 5 near the gas flow channel under various working conditions, which indicates that the motor noise is one of the main noise sources of the air compressor, but that the aerodynamic noise is more prominent. Measuring Point 5 has the highest sound pressure level under different working conditions, which is the result of the joint action of electromagnetic noise and aerodynamic noise.

**Table 4.** Sound pressure level at measuring points [dBA].

| Points | Positions | Idle Condition | Common Condition | Rated Condition |
|---|---|---|---|---|
| 1 | Suction pipeline | 79.7 | 82.9 | 89.1 |
| 2 | First stage volute | 80.3 | 84.8 | 91.1 |
| 3 | Near motor | 78.2 | 82.5 | 88.3 |
| 4 | Outlet of first stage volute | 80.0 | 81.4 | 87.3 |
| 5 | Inter-stage pipeline | 82.8 | 87.0 | 92.2 |
| 6 | Elbow at second stage inlet | 77.5 | 82.3 | 89.5 |
| 7 | Second stage volute | 74.9 | 82.1 | 87.1 |
| 8 | Discharge pipeline | 76.9 | 78.7 | 85.8 |
| 9 | 1 m from the compressor in the horizontal direction | 60.6 | 65.1 | 78.8 |

Figure 5 shows the near-field noise spectrum characteristics of Measuring Point 5 on the surface of the compressor under different working conditions. The noise of the compressor has significant periodic harmonic characteristics. When the compressor is working under the common condition, the noise is prominent in the fundamental frequency (1000 Hz) and the second harmonic frequency (2000 Hz) of the rotation speed, with the noise peak at 76.9 dBA and 83.8 dBA, respectively. The noise peak of the high-order rotation frequency shows a downward trend with the increase in frequency. The discrete single-tone noise of the blade passing frequency (15,000 Hz) is not prominent, and its sound pressure level is only 30.6 dBA, which is far lower than the sound pressure level of the rotation fundamental frequency and its second harmonic frequency. The test results of the noise characteristics are different from the conclusions of previous studies [2–9]. The rotation speed of the centrifugal air compressor for vehicle fuel cells is higher and the number of blades is more. The wavelength at the blade passing frequency is longer, and the attenuation of sound energy in the propagation process is larger. As a result, the noise pressure level at the blade passing frequency is far lower than the noise pressure level at the blade rotation fundamental frequency. The change in rotation speed also has an impact on the noise peak and the overall noise sound pressure level. The noise peak at the rotating fundamental frequency and the low order harmonic frequency gradually increases with the increase in rotation speed, and the broadband noise sound pressure level is higher.

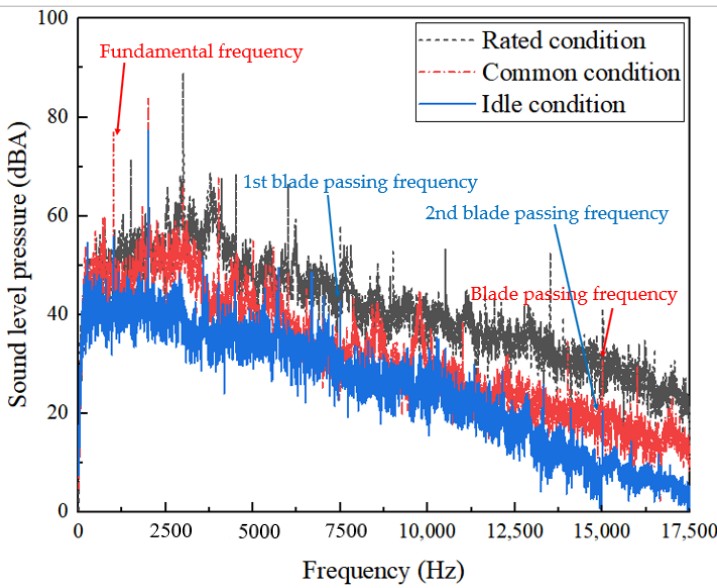

**Figure 5.** Near-field noise spectrum characteristics under different working conditions.

### 2.2. Noise Sources Identification

The high-speed centrifugal air compressor for vehicle fuel cells is a compact multi-noise source device. The electromagnetic, aerodynamic, and structural vibration noises have their own characteristic frequencies. The acoustic characteristics transmitted to different measuring points are different and have relatively independent characteristics. The separation of acoustic signals by the ICA [13,14] (independent component analysis) method based on negative entropy maximization can identify the sound source of the air compressor and clarify the direction for noise improvement.

As shown in Figure 6, the independent component Y2 and the independent component Y3 have typical discrete characteristic frequencies. The signal peak of Y3 occurs at the fundamental frequency, the second frequency, and the third frequency of the rotation frequency, and reaches the maximum at the second frequency. According to the research of Han Xueyan [15] and others, the radial electromagnetic force in the motor acts on the stator core, causing the stator and casing of the motor to vibrate periodically, which is the main reason for the electromagnetic noise generated by the motor. The maximum amplitude of the radial electromagnetic force pulsation of the 2-pole 24-cogged PMSM (permanent magnet synchronous motor) used in the high-speed centrifugal air compressor in this paper appears at the second harmonic frequency of the rotation frequency [16]. Therefore, the radiated acoustic signal is prominent at the second harmonic frequency of the rotation frequency and attenuates sharply at the fundamental frequency and the third harmonic frequency. Therefore, Y3 is the electromagnetic noise source.

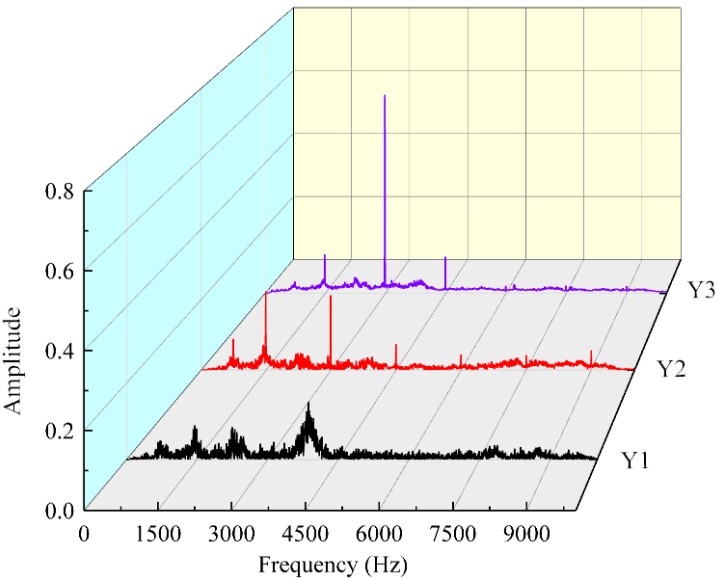

**Figure 6.** Frequency characteristics of independent components.

The maximum amplitude of the independent component Y2 appears at the rotation fundamental frequency and, with the increase in frequency, the signal peak of the characteristic frequency gradually decreases. Therefore, Y2 is the "buzz-saw" noise caused by the periodic rotation of the impeller, including the aerodynamic noise and mechanical noise. The independent component Y1 is concentrated in the low-frequency band, especially the narrow-band noise near 3866 Hz, without obvious periodic characteristics. Therefore, the independent component Y1 signal is the wide-band noise caused by the secondary flow and flow separation process of gas in the blade tip clearance [11]. Based on the noise characteristics of the high-speed centrifugal air compressor of the vehicle fuel cell, the noise signal of the rotating fundamental frequency is mainly composed of the fundamental frequency signal of the "buzz-saw" noise. The second harmonic frequency is composed of the motor noise and the "buzz-saw" noise, forming the maximum amplitude of the noise. The sound pressure levels of the discrete single-tone noise and the wide-band noise are low.

### 3. Characteristics of the Aerodynamic Noise

By establishing the CFD model of the two-stage centrifugal air compressor, its internal gas flow characteristics are studied, and the excitation source of aerodynamic noise is investigated. A CFD–BEM coupled aerodynamic acoustic calculation model is established to study the aerodynamic noise induction mechanism and provide a basis for structural improvement.

*3.1. CFD Method for Exploring the Source of Aerodynamic Noise*

Here, CFD modeling is carried out for the two-stage centrifugal air compressor. The impeller speed and boundary conditions in the model are set according to the operating parameters under rated conditions in Table 2. The inlet boundary is the pressure inlet (101.3 kPa) and the outlet boundary is the pressure outlet (257.0 kPa). The wall surfaces are all non-slip walls. The y+ of the mesh is set to 100, and the wall distance is $3.2 \times 10^{-4}$ m. The mesh in the impeller domain and diffuser domain is the structured hexahedron mesh. The mesh in the volute domain is the unstructured tetrahedral mesh. The minimum elements' quality is above 0.2. The interface models between the moving and stationary parts are the frozen rotor for steady-state simulation and the transient rotor–stator for transient simulation. The boundary conditions for turbulence fractional intensity are 0.05 at both the inlet and the outlet. The Eddy length scales are 0.0386 m and 0.0394 m at the inlet and the outlet, respectively. The CFD calculation adopts ANSYS CFX for the transient solution. The time step is $1.852 \times 10^{-6}$ s, and the corresponding impeller rotates $1°$ at each time step under the rated speed [9]. According to the actual structure of the air compressor, the model is established, as shown in Figure 7.

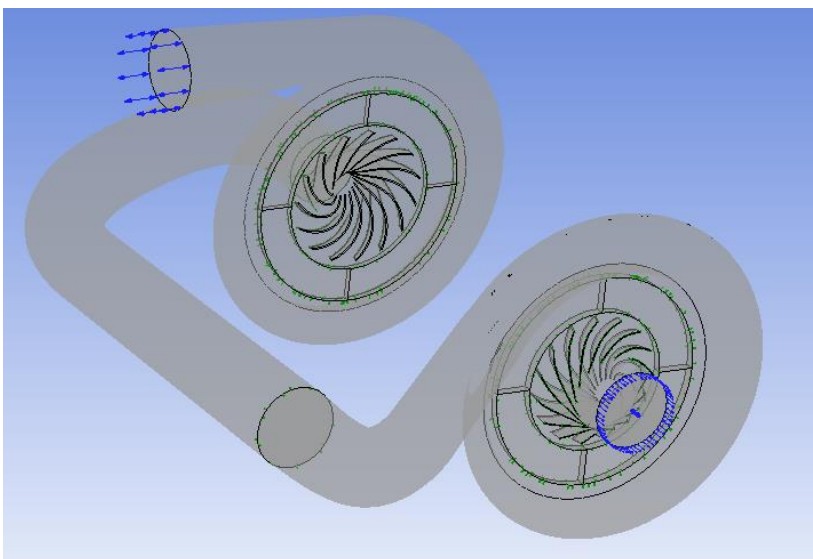

**Figure 7.** CFD model of the compressor.

As the three operating conditions shown in Table 3 are all near the design condition, the Reynold-averaged Navier–Stokes (RANS) equation is used as the governing equation of the model [17], and the shear stress transport (SST) *k-ω* model is used as the turbulence model. Six sets of grids with different element numbers are set for the two-stage centrifugal air compressor, and the mass flow of the air compressor is selected as the basis for judging the grid independence. The curve is shown in Figure 8. It can be seen from the figure that the mass flow calculation difference is less than 0.5% after increasing the number of elements in Case 3, and the number of elements in Case 3 meets the calculation requirements.

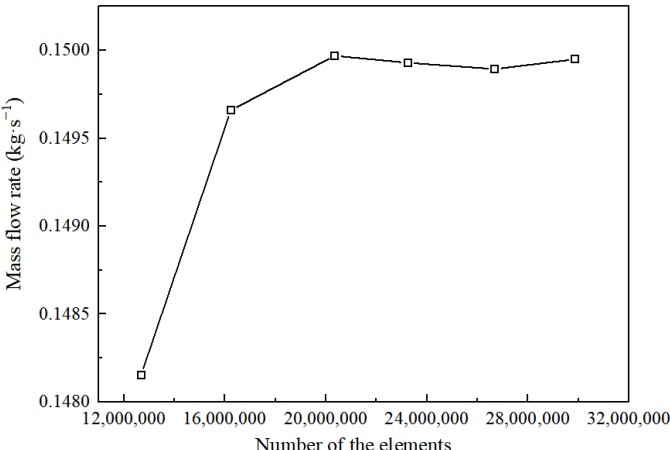

**Figure 8.** Grid independence verification of the CFD model.

The results of CFD calculation are verified by the thermal performance obtained from the test. The pressure ratio (*PR*) of the compressor is for two stages and calculated as follows:

$$PR = P_{out}/P_{in} \tag{1}$$

where $P_{out}$ is the total pressure of the outlet of the second volute and $P_{in}$ is the total pressure of the inlet of the first impeller. As shown in Table 5, the error between the mass flow rate and pressure ratio of the air compressor calculated by CFD and the test results is within 0.5%. As a result, the model calculation is accurate and reliable.

**Table 5.** Error of the CFD results.

|  | **Test Results** | **Calculation Results** | **Error** |
|---|---|---|---|
| Pressure ratio | 2.51 | 2.512 | 0.080% |
| Mass flow rate [g·s$^{-1}$] | 125.36 | 125.687 | 0.26% |

The flow separation will not only bring power loss and efficiency reduction but also induce aerodynamic noise. The change in static entropy can provide a reference for the study of flow separation. Figures 9 and 10 are the B–B static entropy contours of the first and second stage impellers at 90% and 50% blade heights, respectively. There is flow separation caused by the tip clearance flow in both the first and second stage impellers. There is obvious leading-edge flow separation in the second stage impeller. There is the trailing edge flow separation phenomenon in first and second stage impellers, of which the first stage impeller is more violent.

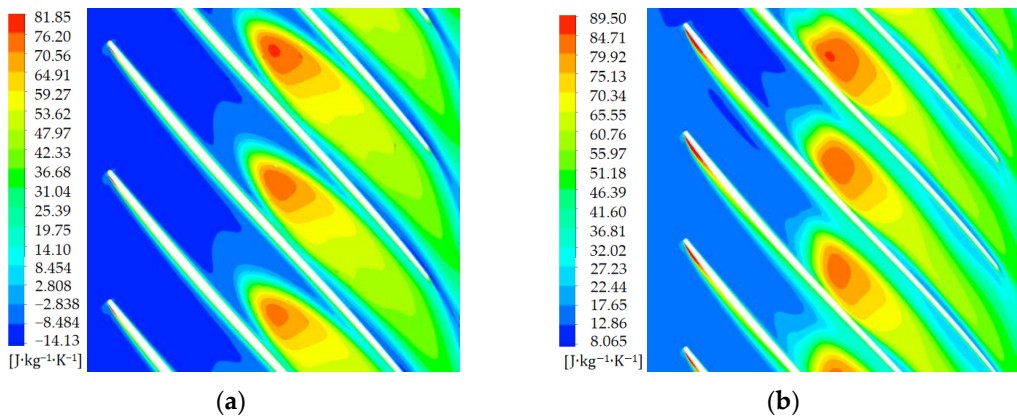

**Figure 9.** Static entropy contours of 90% heights of impellers. (**a**) First stage. (**b**) Second stage.

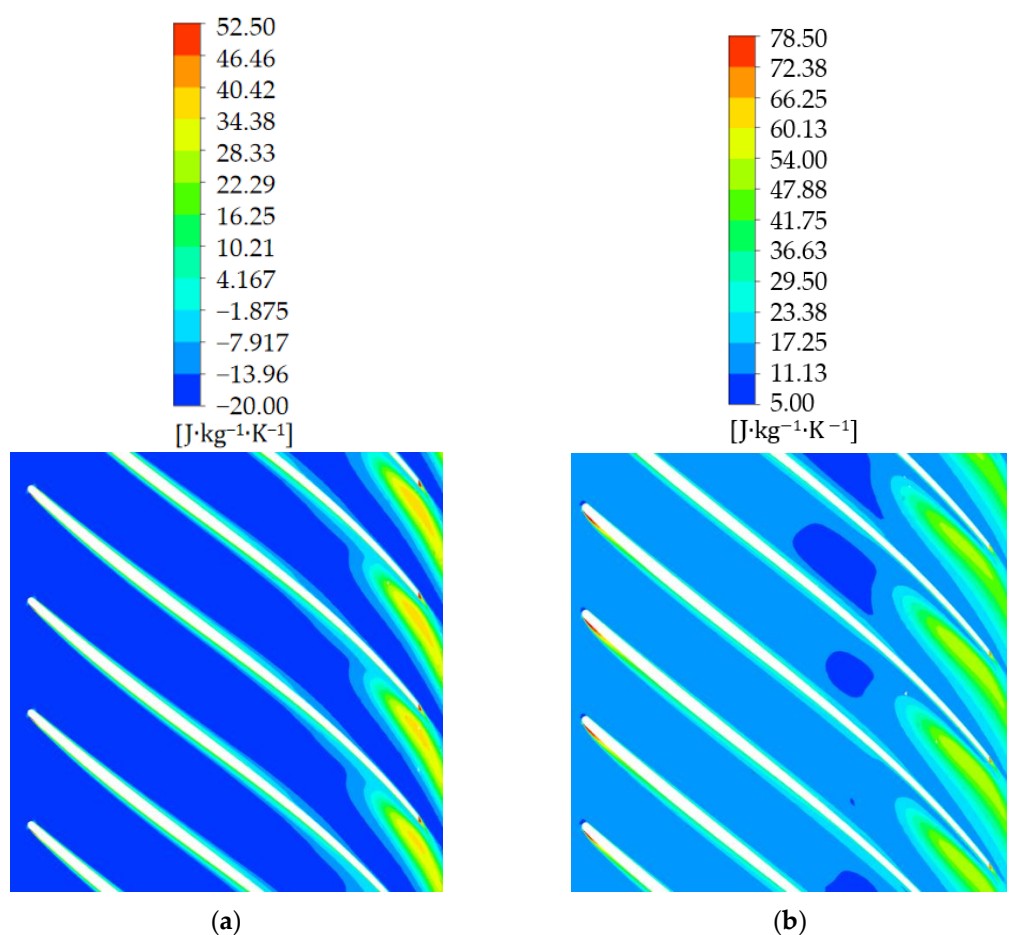

**Figure 10.** Static entropy contours of 50% heights of impellers. (**a**) First stage. (**b**) Second stage.

Figure 11 shows the static pressure contours of the first and second stage diffusers and volutes. The cross-sectional position is at the half width of the diffusers and parallel to the bottom of diffusers. The flow separation at the trailing edge of the second stage impeller brings the obvious uneven pressure at the inlet of the diffuser, and the low-pressure region distributed along the circumference appears.

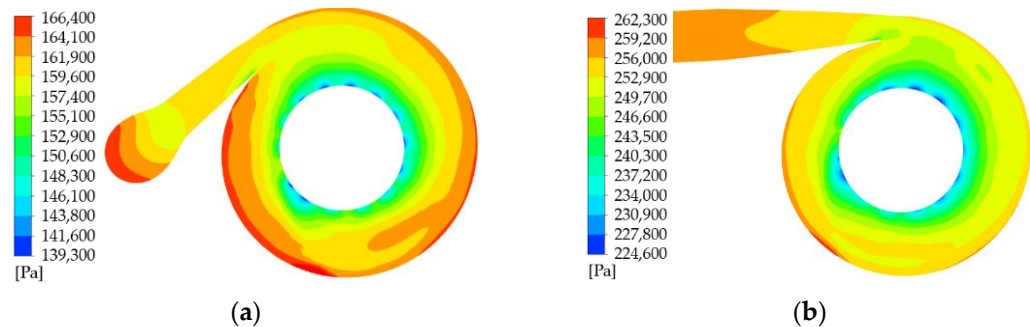

**Figure 11.** Static pressure contours of diffusers and volutes. (**a**) First stage. (**b**) Second stage.

### 3.2. CFD–BEM Coupling Model for Exploring the Inducing Mechanism

A CFD–BEM coupled aerodynamic acoustic calculation model is established to study the aerodynamic noise induction mechanism. Acoustic calculation is used to obtain the solution of the Helmholtz equation satisfying acoustic boundary conditions through the numerical solution [6]. The model is composed of a rotating dipole sound source, the acoustic boundary element mesh, and the field point. In terms of the sound source, the RANS method is applied to calculate the unsteady flow of the compressor, and the time

domain fluctuating pressure on the impeller surface is set as the acoustic excitation source through the boundary conditions for aeroacoustic calculation, as shown in Figure 12. For the flow simulation of the centrifugal compressor, the rotating dipole is the main acoustic source [8]. Thus, in this paper, the sound source is simplified as a dipole. The acoustic boundary element grid is composed of the inner wall of the flow channel formed by the diffuser and volute. The inlet boundary is set as the opening, and the outlet of the volute is set as the non-reflecting boundary. Field points include acoustic monitoring points at the inlet and outlet of the acoustic model and acoustic directional points. Field Points SP1 and SP2 are set at 20 mm away from the volute inlet and outlet to monitor the near-field noise radiated. Acoustic directional field points are set on the plane composed of the inlet and outlet axes. Twelve field points are evenly distributed around the circumference, with the center of the inlet section of the centrifuge as the center and the radius of 1 m to study the sound pressure directivity during the outward radiation of the noise.

According to the time-domain sampling theorem in Nyquist sampling law, the maximum allowable frequency of the acoustic model can be calculated from the time step calculated by the unsteady flow field, which is more than 100 kHz, which is far higher than the maximum audible frequency of 20 kHz, meeting the calculation and analysis requirements. The transient pressure data of 10 cycles on the impeller surface is selected as the acoustic excitation source for acoustic calculation. The highest calculation frequency is 16 times the fundamental frequency of impeller rotation. The calculation range includes the first-order blade passing frequency, and the frequency resolution is 1/10 of the fundamental frequency of impeller rotation. According to the basic assumption of acoustic calculation, we ensure that there are at least six units within the minimum wavelength range, and set the maximum side length of the acoustic unit according to the highest calculation frequency under different operating speeds.

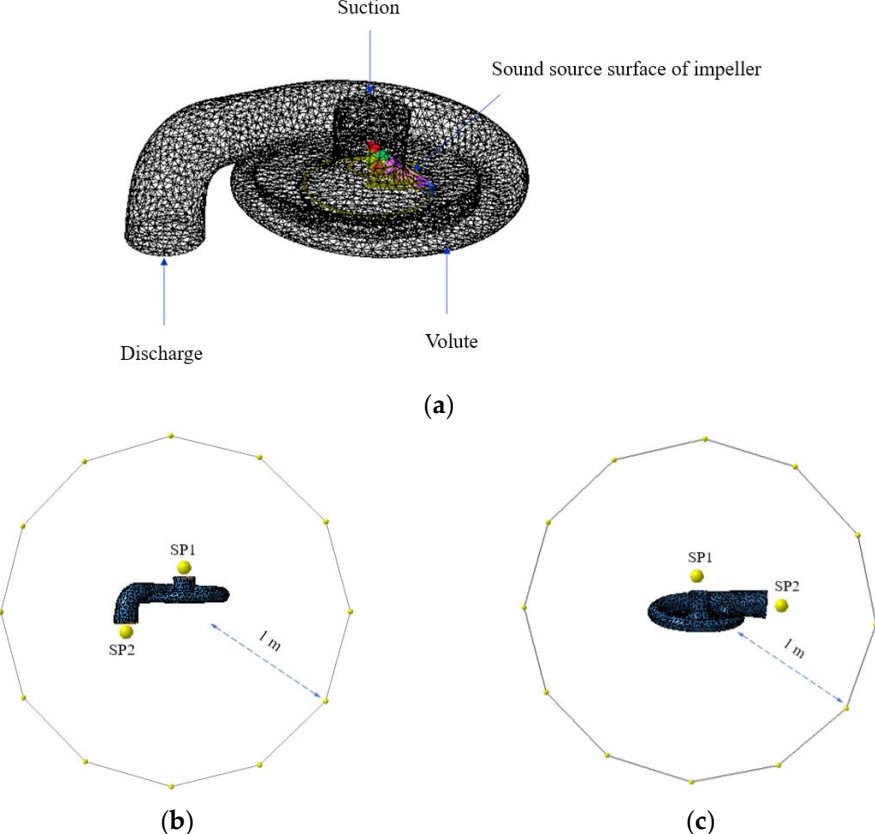

**Figure 12.** Aeroacoustic prediction model. (**a**) Acoustic model of the centrifugal air compressor. (**b**) Field point setting of the first stage volute. (**c**) Field point setting of the second stage volute.

Three sets of grids with different element numbers are set for the aeroacoustic prediction model, and the noise spectrum is selected as the basis for judging the grid independence. The element numbers of the three cases are 1462, 3129, and 6984, respectively. The spectrum is shown in Figure 13. It can be seen from the figure that the noise spectrum fluctuation between Case 2 and Case 3 is small. The number of elements in Case 2 meets the calculation requirements.

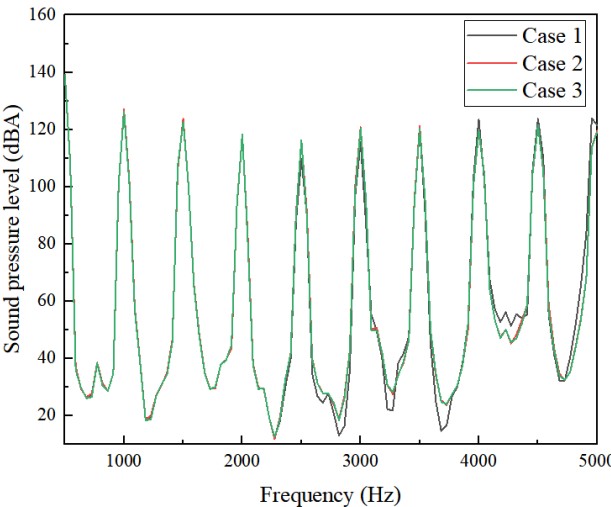

**Figure 13.** Grid independence verification of the CFD–BEM coupling model.

Figure 14 shows the validation of the CFD–BEM coupling model. The calculated aerodynamic noise of the compressor under the rated condition is compared with the separated acoustic signal Y2. As shown in the figure, the calculated spectrum shows good agreement on the frequency characteristics of the "buzz saw noise" especially in the low-frequency range. The differences in the high-frequency range are mainly caused by the higher attenuation in the process of transmission in actual measurement due to the shorter wavelength, which is not considered in the model.

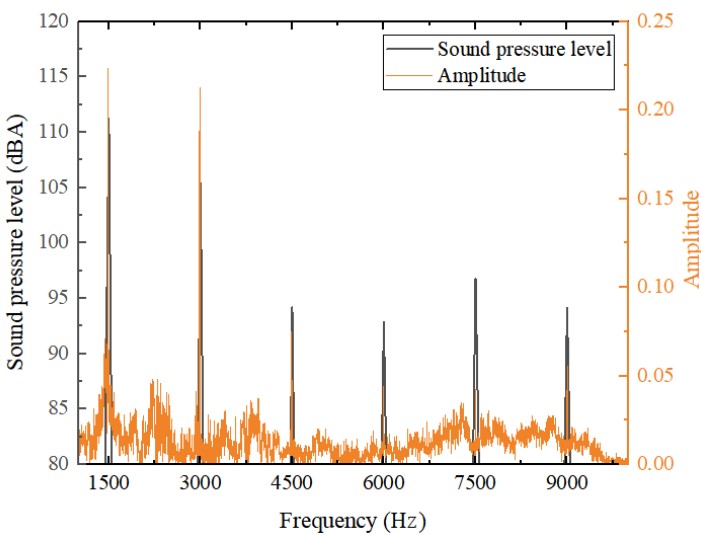

**Figure 14.** Validation of the CFD–BEM coupling model.

Based on the CFD results of the two stage centrifugal air compressor for vehicle fuel cells, the CFD–BEM coupling calculation is carried out to predict the aerodynamic noise characteristics. Figure 15 is the noise spectrum at the near-field Monitoring Points SP1 and SP2 of the first stage and second stage acoustic models of the centrifugal air compressor under the idle condition. The aerodynamic noise spectrum characteristics of Monitoring Points SP1 and

SP2 are highly similar, with significant periodic harmonic characteristics, and the noise peak is prominent. In particular, the "buzz-saw" noise of the rotating fundamental frequency and the second harmonic frequency is significant and dominates, which is the key factor affecting the aerodynamic noise of the centrifugal air compressor.

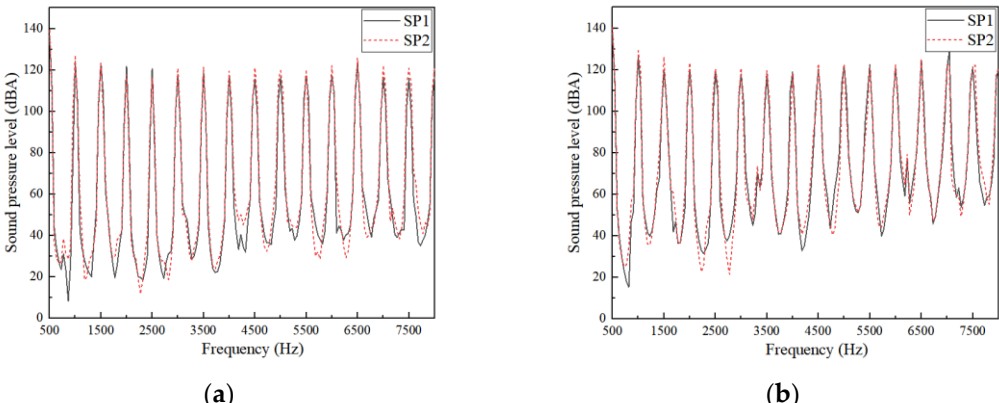

**Figure 15.** Near-field aerodynamic noise characteristics of air compressor at idle condition. (**a**) First stage. (**b**) Second stage.

The sound pressure directivity of the noise radiation of the acoustic models of the first and second stage centrifugal air compressor under different working conditions is shown in Figure 16. The radiation of aerodynamic noise of the acoustic model in the first stage has directivity, and the directivity of sound pressure gradually increases with the increase in rotating speed. The cone-shaped areas of the inlet axis, namely −60–60° and 120–240°, show strong directivity, which is derived from the noise radiated from the inlet and outlet. The noise radiated from the outlet is slightly higher than that of the inlet, indicating that the flow separation at the inlet and outlet of the impeller is the key factor inducing the aerodynamic noise, and the flow separation at the trailing edge has a greater impact. The radiation of the aerodynamic noise of the two-stage acoustic model of the centrifuge also has directionality, which is especially obvious under the rated working condition and the common working condition. The noise value is greater in the area of −60–120° of the inlet axis, which is derived from the outward radiation of the aerodynamic noise of the centrifuge from the inlet and outlet.

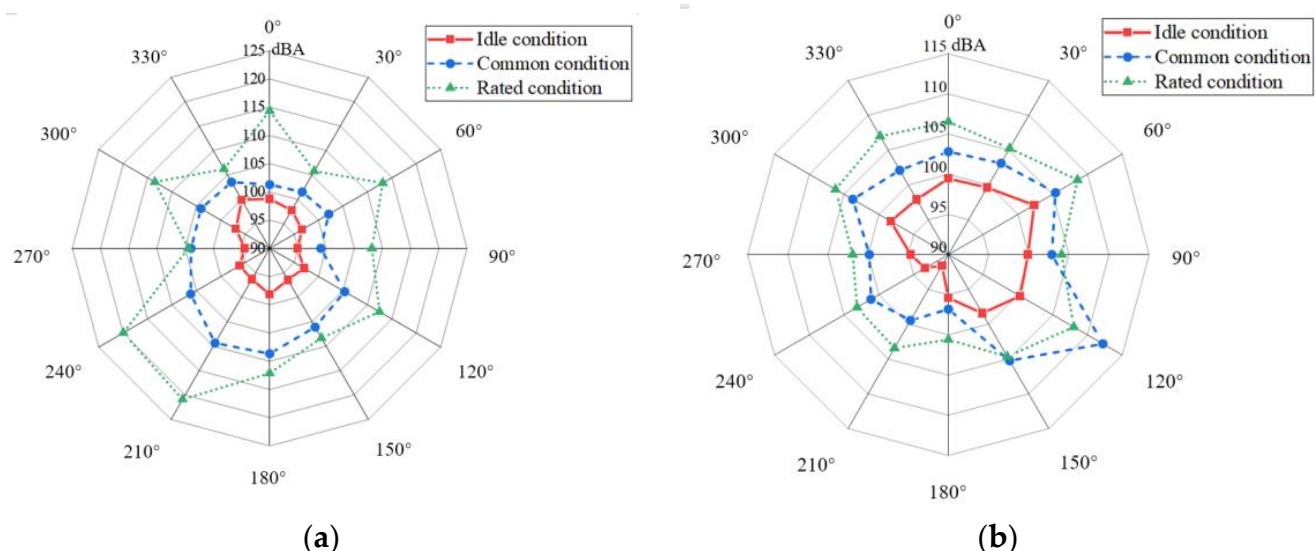

**Figure 16.** Noise radiation directivity of centrifugal impellers. (**a**) First stage. (**b**) Second stage.

According to the flow characteristics and aerodynamic acoustic calculation, the air compressor exhibits the flow separation phenomenon in the process of impeller acceleration, including the flow separation of the secondary flow in the tip clearance, the leading edge, and the trailing edge, which leads to uneven pressure distribution in the diffuser and volute, induces aerodynamic noise, and mainly radiates outward along the outlet direction [18,19]. The noise at the rotating fundamental frequency and its harmonic frequency is particularly significant.

## 4. Noise Control

In view of the flow separation phenomenon found above, the structure of the compressor is improved, so as to suppress the noise from the source. The improved flow characteristics and noise characteristics are calculated and compared, and the improved effect is verified by experiments. Furthermore, the noise propagation process is suppressed by using a perforated muffler in the inter-stage pipeline.

### 4.1. Optimization of Structural Parameters

In view of the above research, in order to reduce the "buzz-saw" noise, on the one hand, based on the flow characteristics and previous studies, the flow separation is reduced by increasing the blade outlet angle [20] and increasing the impeller outlet radius. On the other hand, by increasing the width of the diffuser and the outlet area of the first stage volute, the interference between the rotating flow and the volute tongue is reduced. The schematic of the optimized parameters is shown in Figure 17.

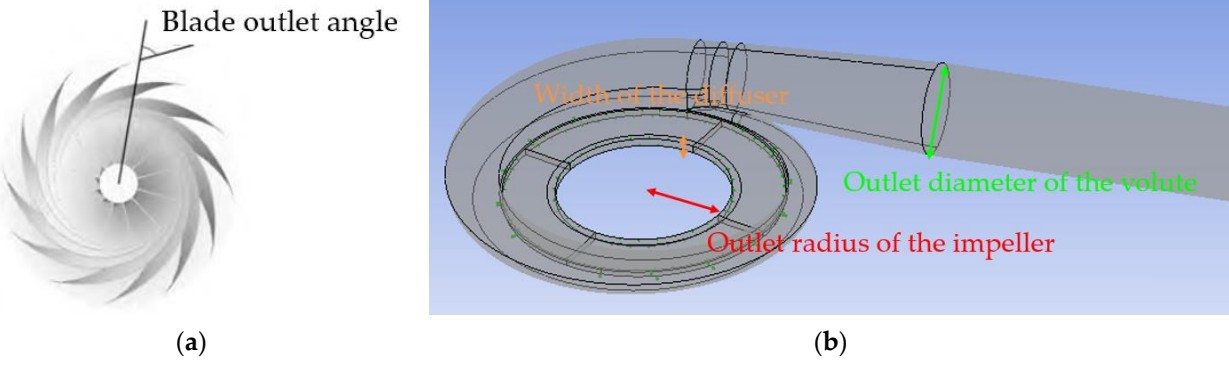

(**a**)  (**b**)

**Figure 17.** Schematic of the optimized parameters. (**a**) Blade outlet angle. (**b**) Other optimized parameters.

The aerodynamic optimization platform established in Reference [21] is adopted for the parameter selection of the air compressor structure improvement in this paper. The optimization platform integrates geometric parameterization, a surrogate model, sensitivity analysis, and a multi-objective genetic algorithm. Among these, the sensitivity analysis mainly adopts the optimal prediction meta-model (MOP), and the optimization process adopts the multi-objective genetic optimization algorithm based on the MOP surrogate model. Based on the optimal design variable subset of each target parameter, different surrogate models are established by the polynomial method, moving least squares method, and Kriging method, respectively. A set of competitive metamodels for each target parameter is formed. The evaluation is performed using a model-independent cross-validation method proposed by Most and Will [22], called the coefficient of prognosis (*CoP*), expressed as follows:

$$CoP = \left( \frac{\mathrm{E}\left[Y_{\text{test}} \cdot \hat{Y}_{\text{test}}\right]}{\sigma_{Y_{\text{test}}} \cdot \sigma_{\hat{Y}_{\text{test}}}} \right)^2 \tag{2}$$

where the value of the *CoP* is obtained based on the test sample point data of cross-validation. The prediction error is used to evaluate the prediction quality of the model. Both regression models and interpolation models are applicable. According to the size of the *CoP* of the meta-model, the optimal prediction meta-model of each target parameter is determined, that is, the MOP surrogate model. The sensitivity of the design parameters

of the optimal prediction metamodel for target parameters is quantified by using the *CoP* value of the optimal prediction metamodel together with the total effect sensitivity index. That is, the variance contribution of a single design parameter is the product of the *CoP* value and the overall effect sensitivity index, as follows:

$$CoP(X_i) = \text{CoP} \cdot S_{\text{T}}^{\text{MOP}}(X_i) \tag{3}$$

$$S_{\text{T}}^{\text{MOP}}(X_i) = 1 - V(Y|X_{\sim i})/V(Y) \tag{4}$$

where $Y$ is the output variable of the model, $X_i$ is the input variable of the model, $V(Y)$ is the variance of $Y$ caused by all input variables, and $V(Y|X_{\sim i})$ is the variance of $Y$ caused by all input variables except $X_i$. Three target parameters are defined for the optimization object of this study, which are the total pressure loss coefficient, the average absolute airflow angle of the impeller outlet section, and the pressure range of the impeller outlet section. The ranges and changes of structural parameters before and after improvement are shown in Table 6.

**Table 6.** Ranges and changes in structural parameters before and after improvement.

| Structural Parameters | Before Improvement | Lower Limit | Upper Limit | After Improvement |
|---|---|---|---|---|
| Blade outlet angle of the first stage impeller [°] | 52.0 | 50 | 54 | 52.5 |
| Blade outlet angle of the second stage impeller [°] | 51.0 | 50 | 54 | 53.4 |
| Outlet radius of the first stage impeller [mm] | 33.1 | 26 | 40 | 37.1 |
| Outlet radius of the second stage impeller [mm] | 33.5 | 26 | 40 | 35.2 |
| Width of the first stage diffuser [mm] | 4.0 | 2 | 6 | 4.9 |
| Width of the second stage diffuser [mm] | 3.0 | 2 | 6 | 3.2 |
| Outlet diameter of the first stage volute [mm] | 36.8 | 30 | 44 | 41.6 |
| Outlet diameter of the second stage volute [mm] | 39.4 | 30 | 46 | 39.4 |

Figures 18 and 19 are B–B static entropy contours of the first and second stage impellers of the improved air compressor at 90% and 50% blade height. Since the blade tip clearance height is the same, the flow separation phenomenon in the blade tip clearance is not significantly improved, but the flow separation phenomenon at the leading edge and the trailing edge is significantly improved. Figure 20 is a pressure contour of the first and second stage diffusers and volutes at 50% of the width of the diffuser after the improvement. After the improvement, the pressure unevenness at the inlet of the diffuser has been significantly improved, and the static pressure gradient at the volute tongue is smaller.

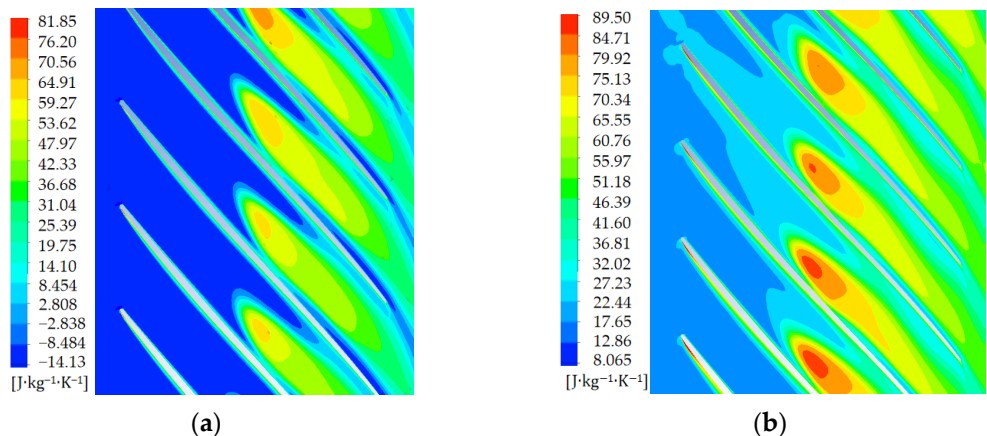

**Figure 18.** Static entropy contours of 90% heights of impellers. (**a**) First stage. (**b**) Second stage.

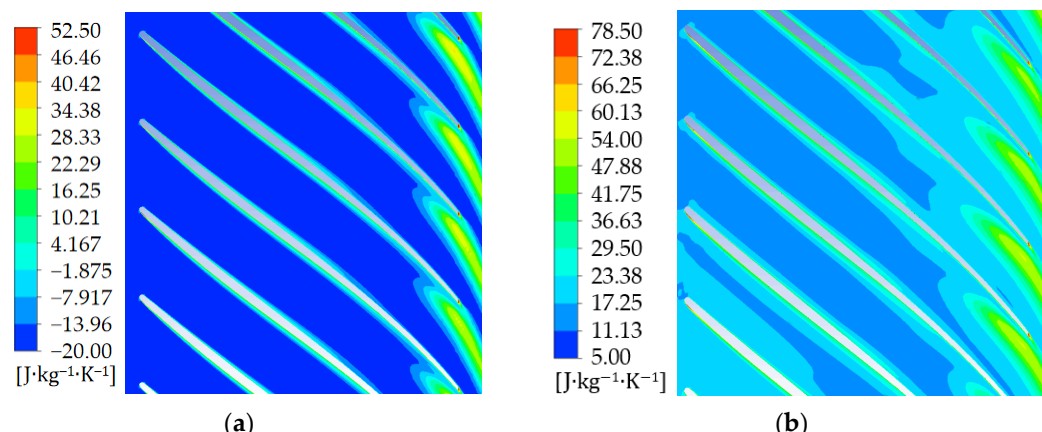

**Figure 19.** Static entropy contours of 50% heights of impellers. (**a**) First stage. (**b**) Second stage.

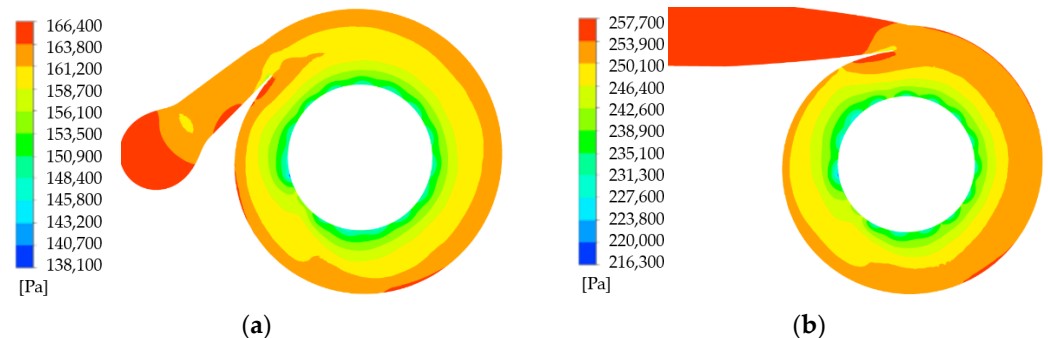

**Figure 20.** Static pressure contours of diffusers and volutes after improvement. (**a**) First stage. (**b**) Second stage.

The aerodynamic acoustic analysis of the improved air compressor was carried out. After the centrifugal impeller is improved, the aerodynamic noise improvement results at the near-field Monitoring Points SP1 and SP2 under the idle condition are shown in Figure 21.

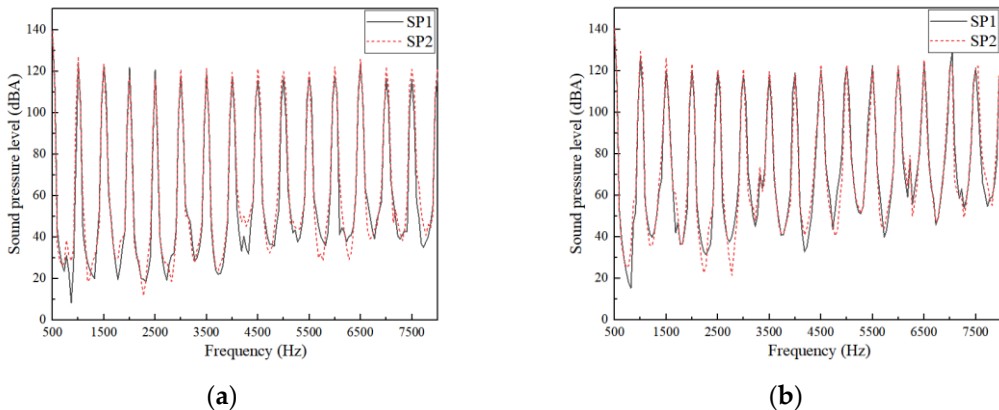

**Figure 21.** Noise spectrum before and after improvement under idle condition. (**a**) SP1. (**b**) SP2.

After the improvement of the structure, the "buzz-saw" noise at the rotating fundamental frequency and its doubling frequency noise are improved. Tables 7–9 lists the noise improvement effects of the rotating fundamental frequency and the doubling frequency of the impeller under different working conditions in detail.

**Table 7.** Noise at rotating fundamental frequency and second harmonic at idle condition [dBA].

| | Fundamental Frequency | | | Second Harmonic | | |
|---|---|---|---|---|---|---|
| | Before Improvement | After Improvement | Difference | Before Improvement | After Improvement | Difference |
| Stage I SP1 | 136.4 | 129.2 | 7.2 | 124.1 | 115.7 | 8.4 |
| Stage I SP2 | 139.2 | 131.2 | 7.9 | 127.1 | 116.9 | 10.2 |
| Stage II SP1 | 136.8 | 128.0 | 8.8 | 126.1 | 117.2 | 8.9 |
| Stage II SP2 | 140.4 | 130.1 | 10.3 | 129.1 | 120.5 | 8.6 |

**Table 8.** Noise at rotating fundamental frequency and second harmonic at common condition [dBA].

| | Fundamental Frequency | | | Second Harmonic | | |
|---|---|---|---|---|---|---|
| | Before Improvement | After Improvement | Difference | Before Improvement | After Improvement | Difference |
| Stage I SP1 | 135.6 | 126.8 | 8.8 | 127.1 | 116.7 | 10.3 |
| Stage I SP2 | 140.7 | 131.0 | 9.7 | 129.1 | 119.2 | 10.0 |
| Stage II SP1 | 135.9 | 127.0 | 8.9 | 128.0 | 121.0 | 7.0 |
| Stage II SP2 | 141.4 | 133.4 | 8.0 | 133.5 | 121.6 | 11.9 |

**Table 9.** Noise at rotating fundamental frequency and second harmonic at rated condition [dBA].

| | Fundamental Frequency | | | Second Harmonic | | |
|---|---|---|---|---|---|---|
| | Before Improvement | After Improvement | Difference | Before Improvement | After Improvement | Difference |
| Stage I SP1 | 136.7 | 127.7 | 9.0 | 127.6 | 115.0 | 12.6 |
| Stage I SP2 | 141.2 | 130.0 | 11.2 | 130.6 | 122.2 | 8.4 |
| Stage II SP1 | 137.9 | 127.3 | 10.6 | 129.5 | 120.9 | 8.6 |
| Stage II SP2 | 142.8 | 135.4 | 7.3 | 133.5 | 123.0 | 10.6 |

After the structure of the centrifuge is improved, the aerodynamic noise of the compressor at the fundamental frequency and the second harmonic frequency of the impeller rotation is improved under different operating conditions, and the difference reaches 12.6 dBA.

*4.2. Experimental Verification on the Improvement of Structure*

The thermal performance and noise characteristics of the improved air compressor were tested. Table 10 shows the comparison of the thermal performance of the air compressor under rated working conditions before and after the improvement. The isentropic efficiency is calculated as follows:

$$\eta_{is} = \frac{PR^{\frac{k-1}{k}} - 1}{\frac{T_{out}}{T_{in}} - 1} \tag{5}$$

where $k$ is the specific heat ratio of air, $T_{out}$ is the temperature of the outlet of the second volute, and $T_{in}$ is the temperature of the inlet of the first impeller. After the improvement, the performance of the air compressor changed little, the pressure ratio decreased by 0.40%, the mass flow decreased by 0.19%, and the isentropic efficiency increased by 0.62%. From the comparison results, the thermal performance of the air compressor remains at the same level after the improvement of the air compressor structure for the aerodynamic noise.

**Table 10.** Thermal performance before and after improvement.

| Performances | Before Improvement | After Improvement |
| --- | --- | --- |
| Pressure ratio | 2.51 | 2.50 |
| Mass flow rate [g·s$^{-1}$] | 125.36 | 125.12 |
| Isentropic efficiency | 77.29% | 77.77% |

The comparison results of near-surface noise of the air compressor under rated working conditions are shown in Table 11. The noise at the first and second stage volutes is reduced by 5.3 dBA and 3.3 dBA, respectively, and the noise at the measuring point of the inter-stage pipeline is reduced by 7.4 dBA.

**Table 11.** Sound pressure level before and after improvement [dBA].

| Points | Positions | Before Improvement | After Improvement |
| --- | --- | --- | --- |
| 2 | First stage volute | 91.1 | 85.8 |
| 5 | Inter-stage pipeline | 92.2 | 84.8 |
| 7 | Second stage volute | 87.1 | 83.8 |
| 9 | 1 m from the compressor in the horizontal direction | 78.8 | 74.7 |

Figure 22 shows the noise characteristics at Point 7 before and after improvement. After the improvement, the peak of the sound pressure level at the fundamental frequency and second harmonic is significantly reduced. There is a spill-over effect in the after improvement spectrum at 6000 Hz. Reducing the noise at the target frequency may cause a slight increase in the noise at other ranges of frequency. The possible reason for this phenomenon is that, in the process of converting a large-scale eddy to a small-scale eddy, the frequency of the noise induced by eddy will increase.

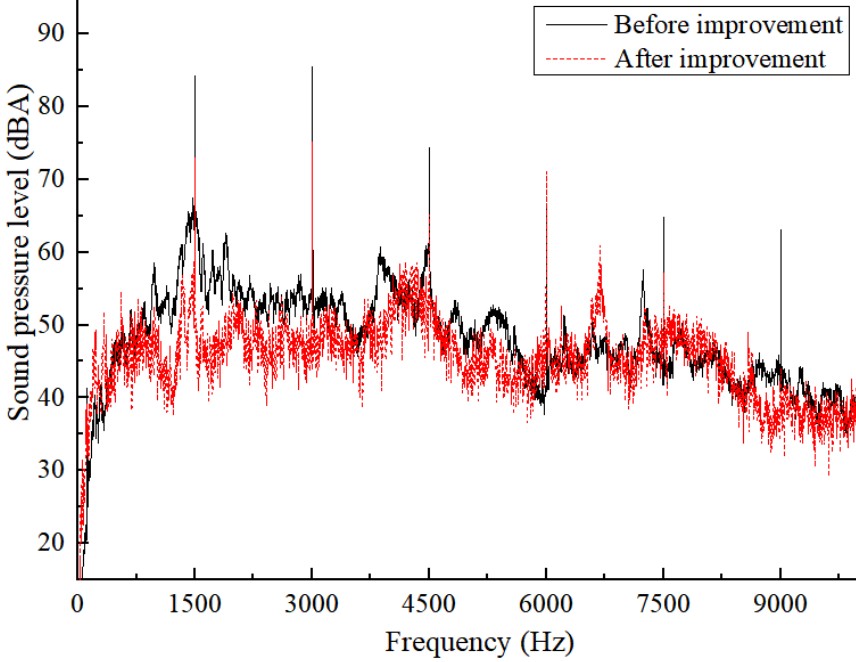

**Figure 22.** Noise characteristics before and after improvement.

### 4.3. Application of the Perforated Muffler

The vibration of the pipeline and shell caused by flow pulsation is the direct factor of aerodynamic noise. Suppressing the transmission of the pulsation can effectively reduce aerodynamic noise. Although the resistive muffler and the impedance composite muffler can broaden the noise reduction frequency range, they cannot be applied in the centrifugal air compressor for vehicle fuel cells because the resistive material will pollute the air quality and increase the pressure loss of the system. Therefore, based on the noise characteristics of the air compressor, a broadband low-resistance perforated muffler is proposed, which has the advantages of wide reduction frequency, an obvious noise reduction effect, small pressure loss, and the fact that it is pollution-free. The schematic diagram of the perforated muffler is shown in Figure 23.

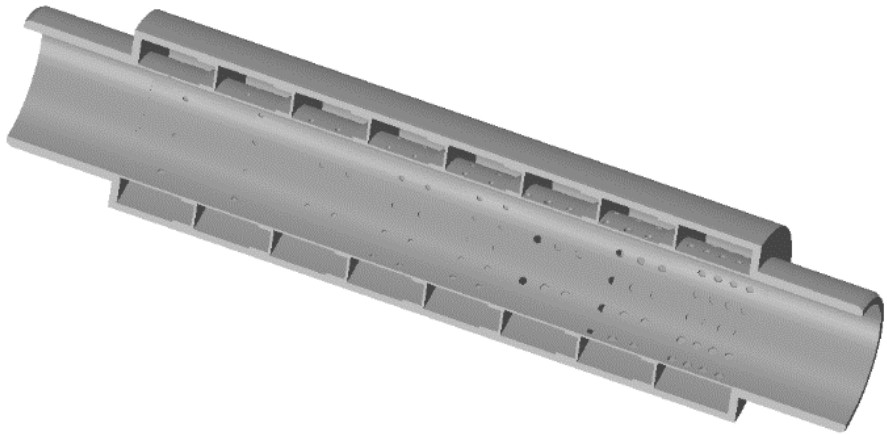

**Figure 23.** Schematic diagram of the perforated muffler.

The perforated muffler is designed based on the Helmholtz principle. The perforation on the perforated tube and the resonance cavity behind the perforated tube form a Helmholtz resonance cavity [23]. When the pulsation frequency of the airflow is close to the natural frequency of the resonant cavity, the resonant cavity can absorb the acoustic energy and reduce the amplitude of the pulsation. The muffler adopts a multi-cavity series connection, which can be regarded as the series connection of many Helmholtz resonators, effectively broadening the range of the reduction frequency. The flow chart of the design for the muffler is shown in Figure 24. The general design principle is to maximize the resonant cavity within the confined compressor structure since a larger resonant cavity has a better noise reduction effect. Therefore, according to the structure of compressor, the geometric parameter is first selected to achieve a maximum cavity. Then, in order to ensure the structural strength and reliability, the thickness of the perforated wall is consistent with the thickness of the original structure. Afterwards, the noise reduction frequency is calculated and fine-tuned by adjusting the design parameters to meet the targeted noise reduction frequency. In the optimization process, the parameters of the proposed muffler are continuously adjusted, and the attenuation performance of the muffler is evaluated using 3D simulation. When the attenuation frequency is consistent with the targeted attenuation frequency, which is determined by the compressor operation characteristics, the attenuation pulsation is considered as maximum pressure pulsation attenuation, and the selected parameters are determined. Since the muffler is designed according to the specific structure of the compressor, one of the design constraints is the compressor structure, which limits the dimension and shape of the mufflers. Another design constraint is the compressor operating condition, which determines the specific frequency or specific frequency band.

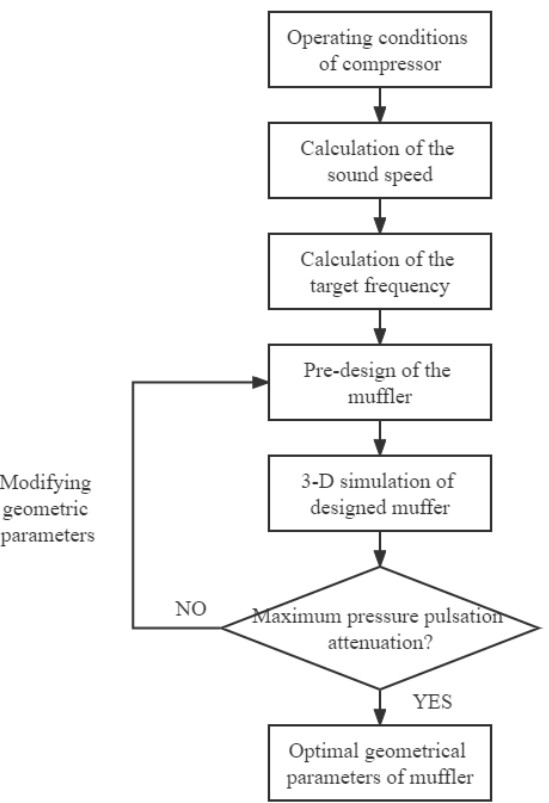

**Figure 24.** Flow chart of the design method for the muffler.

The transmission loss of the perforated muffler is shown in Figure 25. The muffler has significant broadband characteristics, and the noise reduction effect is above 10 dBA in the frequency range of 1000–4000 Hz, which can effectively reduce the aerodynamic noise of the centrifugal air compressor.

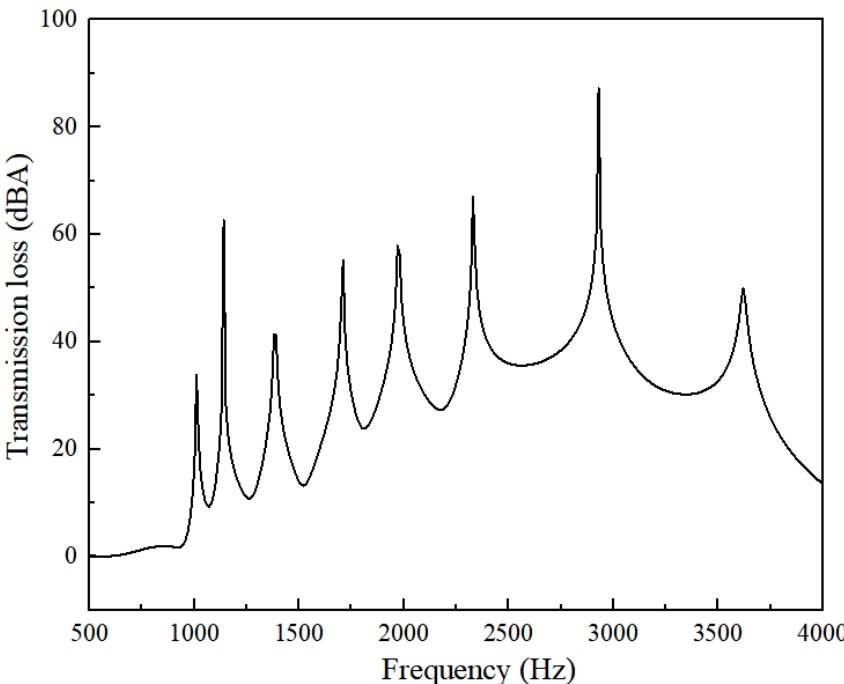

**Figure 25.** Transmission loss of the perforated muffler.

Figure 26 shows the tested noise spectrum of the compressor under the rated condition at Measuring Point 9 before and after the application of the perforated muffler. The application of the perforated muffler has an obvious noise attenuation effect on broadband noise and "buzz-saw" noise in the range of 0–4000 Hz. The sound pressure level reduces from 74.7 dBA to 68.9 dBA, by 5.8 dBA.

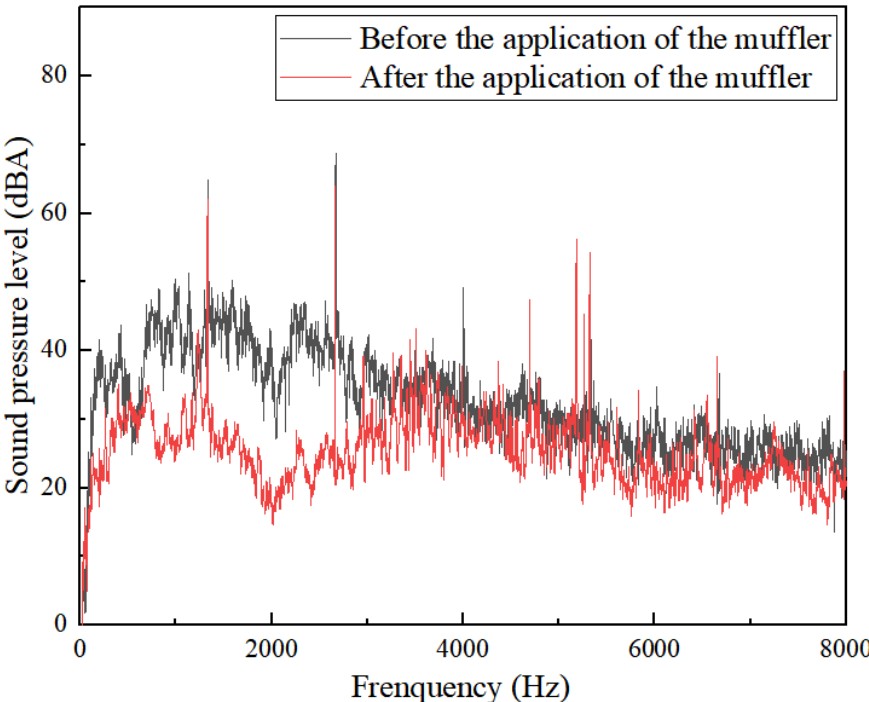

**Figure 26.** Noise spectrum before and after the application of the perforated muffler.

## 5. Conclusions

In order to reduce the noise level of the developed high-speed two-stage centrifugal air compressor for vehicle fuel cells, the experimentally obtained near-field noise characteristics are used to identify the sound source by the independent component analysis method, and the CFD–BEM coupling model is further applied for acoustic calculation and analysis. Based on the analysis results, the structure of the compressor is optimized, and a perforated muffler is applied in the pipe system for the reduction in the noise. The main conclusions are as follows:

(1) The "buzz-saw" noise at the rotating fundamental frequency and its low order harmonic frequency in the aerodynamic noise is prominent in the noise spectrum, while the sound pressure level of the discrete single-tone noise is relatively low;

(2) There are flow separation phenomena in the process of impeller acceleration in the compressor, including the secondary flow in the blade tip clearance, and the flow separation at the leading edge and the trailing edge, which ultimately leads to uneven pressure distribution in diffusers and volutes. In order to reduce the "buzz-saw" noise of the air compressor, the blade outlet angles, the radius of the impeller outlets, the diffuser width, and the outlet area of the first stage volute are increased, so as to reduce the flow separation phenomenon and the interference within the volute tongue;

(3) After the structure optimization, the thermal performance of the air compressor maintained the same level, and the noise level was significantly reduced. The near-field noise at the first and second stage volutes is reduced by 5.3 dBA and 3.3 dBA, respectively, and the near-field noise at the inter-stage pipeline is reduced by 7.4 dBA;

(4) The application of the perforated muffler has an obvious noise attenuation effect on broadband noise and "buzz-saw" noise in the range of 0–4000 Hz. The sound pressure

level reduces from 74.7 dBA to 68.9 dBA, specifically by 5.8 dBA, under the rated condition at 1 m from the compressor in the horizontal direction.

The research in this paper realizes the key technologies of the whole chain from sound source identification and induced mechanism exploration to noise control. The results provide the acoustic theoretical support and a practical basis for the forward development and upgrading of the air compressor for vehicle fuel cells.

**Author Contributions:** Investigation, S.S.; Methodology, S.S., W.C. and M.Z.; Project administration, Z.X.; Software, M.Z.; Supervision, Z.X.; Validation, C.W. and H.C.; Visualization, S.S.; Writing—original draft, S.S.; Writing—review & editing, W.C. and C.W. All authors have read and agreed to the published version of the manuscript.

**Funding:** This research was funded by the National Key Research and Development Program of China, grant number 2019YFB1504604.

**Institutional Review Board Statement:** Not applicable.

**Informed Consent Statement:** Not applicable.

**Data Availability Statement:** Not applicable.

**Acknowledgments:** The support from Gree Electric Appliances Inc and HPC Platform, Xi'an Jiaotong University is appreciated.

**Conflicts of Interest:** The authors declare no conflict of interest.

## Nomenclature

| | | | |
|---|---|---|---|
| CFD | Computational fluid dynamics | BEM | Boundary element method |
| ICA | Independent component analysis | RANS | Reynold-averaged Navier-Stokes |
| SST | Shear stress transport | *PR* | Pressure ratio |
| *p* | Pressure | $\eta_{is}$ | Isentropic efficiency |
| *k* | Specific heat ratio | *T* | Temperature |
| CoP | Coefficient of prognosis | MOP | Optimal prediction meta model |
| *Y* | Output variable | *X* | Input variable |
| *V* | Variance | | |

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
