# Peer review of "Study on Characteristics and Control of Aerodynamic Noise of a High-Speed Centrifugal Air Compressor for Vehicle Fuel Cells"

_applsci, doi:10.3390/app12199962_

Round 1

Reviewer 1 Report

The authors investigate the acoustic performance of a centrifugal compressor experimentally and numerically. The flow is simulated applying a RANS turbulence model adn exrapolated by a beam element method.

One of the main drawbacks is that the experimental and numerical investigation are not compared. There is no validation performed. No similarities or differences in the results discussed.

The importance of the investigation is the acoustic noise. Thus, checking the grid refinement for the mass flow rate feels troublesome. Rather spectra obtained with the different meshes shall be compared. This shows until which frequency the method captures the fluctuations.

The numerical method is often just compared to itself. The improvement is judged very supervicially without frequency weighting.

For noise simulations, special non-reflecting boundary conditions are needed. Nothing is stated about the treatment of acoustic waves at boundaries. - Not boundary conditions are provided for turbulence cuantities.

Please reformulate the conclusion section. Several sentenses are not understandable, e.g. "The noise improvement scheme is proposed".

Finally, it is very difficult to understand what is actually new in the presented work and what is the knowledge gained. Very little literature is cited and compared to. There are many other works that should be referenced;

* Sundstrom et al. (2018) "Acoustic signature of flow instabilities in radial compressors"

* Sharma et al. (2020) "Evaluation of modelling parameters for computing flow-induced noise in a small high-speed centrifugal compressor"

* Freidhager et al. (2021) "Lighthill's analogy applied to an automotive turbocharger compressor"

* Sundstrom et al. (2014) "Assessment of the 3d flow in a centrifugal compressor using steady-state and unsteady flow solvers"

Author Response

Dear reviewer,

Thank you for your letter and comments concerning our manuscript ID applsci-1921716 entitled " Study on Characteristics and Control of Aerodynamic Noise of a High-speed Centrifugal Air Compressor for Vehicle Fuel Cells". Those comments are very helpful for improving our paper. We have studied these comments carefully and also have tried our best to make corrections. The main corrections in the paper and the responses to the reviewer’s comments are as follows:

Comment 1: The authors investigate the acoustic performance of a centrifugal compressor experimentally and numerically. The flow is simulated applying a RANS turbulence model adn exrapolated by a beam element method.

One of the main drawbacks is that the experimental and numerical investigation are not compared. There is no validation performed. No similarities or differences in the results discussed.

Response:

Thank you for your comments and suggestions. We have supplemented the validation of the CFD-BEM coupling model by the comparison of the calculated aerodynamic noise with the separated acoustic signal Y2. The similarities and differences are discussed. Please see the detail in the text as follows:

“Figure 10 shows the validation of the CFD-BEM coupling model. The calculated aerodynamic noise of the compressor under the rated condition is compared with the separated acoustic signal Y2. As shown in the figure, the calculated spectrum shows good agreement on the frequency characteristics of the “buzz saw noise” especially in the low-frequency range. The differences in the high-frequency range are mainly caused by the higher attenuation in the process of transmission in actual measurement due to the shorter wavelength, which is not considered in the model.

Figure 10. Validation of the CFD-BEM coupling model”

Comment 2: The importance of the investigation is the acoustic noise. Thus, checking the grid refinement for the mass flow rate feels troublesome. Rather spectra obtained with the different meshes shall be compared. This shows until which frequency the method captures the fluctuations.

Response:

Thank you for your kind suggestion. Since the characteristics of flow and noise are both targets of research in the paper, we keep the verification of the grid independence of the CFD model and supplement the verification of the grid independence of the CFD-BEM coupling model. Please see the detail in the text as follows:

“Three sets of grids with different element numbers are set for the aeroacoustic prediction model, and the noise spectrum is selected as the basis for judging the grid independence. The element numbers of the three cases are 1462, 3129, and 6984 respectively. The spectrum is shown in Figure 10. It can be seen from the figure that the noise spectrum fluctuation between Case 2 and Case 3 is small. The number of elements in case 2 meets the calculation requirements.

Figure 10. Grid independence verification of the CFD-BEM coupling model”

Comment 3: The numerical method is often just compared to itself. The improvement is judged very supervicially without frequency weighting.

Response:

Thank you for your suggestion. Considering the different sensitivity of human ears to the noise of different frequencies, the measured sound pressure level adopts A weighting in this paper. All the calculated sound pressure levels also adopt A weighting consistent with the test results. The unified weighting method ensures the effectiveness of the judge for noise improvement.

Comment 4: For noise simulations, special non-reflecting boundary conditions are needed. Nothing is stated about the treatment of acoustic waves at boundaries. - Not boundary conditions are provided for turbulence cuantities.

Response:

We would like to appreciate your comment. We have supplemented the introduction of the CFD model in detail. The boundary conditions for turbulence fractional intensity are 0.05 both at the inlet and the outlet. The Eddy length scales are 0.0386 m and 0.0394 m at the inlet and the outlet respectively. The introduction of the CFD-BEM model has been reformulated. Please see the detail in the text as follows:

“A CFD-BEM coupled aerodynamic acoustic calculation model is established to study the aerodynamic noise induction mechanism. The model is composed of a rotating dipole sound source, the acoustic boundary element mesh, and the field point. In terms of the sound source, the RANS method is applied to calculate the unsteady flow of the compressor, and the time domain fluctuating pressure on the impeller surface is set as the acoustic excitation source through the boundary conditions for aeroacoustic calculation. For the flow simulation of the centrifugal compressor, the rotating dipole is the main acoustic source[8], so in this paper, the sound source is simplified as a dipole. The acoustic boundary element grid is composed of the inner wall of the flow channel formed by the diffuser and volute. The inlet boundary is set as the opening and the outlet of the volute is set as the non-reflecting boundary. Field points include acoustic monitoring points at the inlet and outlet of the acoustic model and acoustic directional points. Field points SP1 and SP2 are set at 20 mm away from the volute inlet and outlet to monitor the near-field noise radiated. Acoustic directional field points are set on the plane composed of the inlet and outlet axes. 12 field points are evenly distributed on the circumference with the center of the inlet section of the centrifuge as the center and the radius of 1 m to study the sound pressure directivity during the outward radiation of the noise.”

Comment 5: Please reformulate the conclusion section. Several sentenses are not understandable, e.g. "The noise improvement scheme is proposed".

Response:

We would like to appreciate your suggestion. We have reformulated the conclusion section. Please see the detail in the text as follows:

“In order to reduce the noise level of the developed high-speed two-stage centrifugal air compressor for vehicle fuel cells, the obtained near-field noise characteristics experimentally are used to identify the sound source by the independent component analysis method, and the CFD-BEM coupling model is further applied for acoustic calculation and analysis. Based on the analysis results, the structure of the compressor is optimized and a perforated muffler is applied in the pipe system for the reduction of the noise. The main conclusions are as follows:

(1) The “buzz-saw” noise at the rotating fundamental frequency and its low order harmonic frequency is prominent in the noise spectrum, while the sound pressure level of the discrete single-tone noise is relatively low.

(2) There are flow separation phenomena in the process of impeller acceleration in the compressor, including the secondary flow in the blade tip clearance, and the flow separation at the leading edge and the trailing edge, which ultimately leads to uneven pressure distribution in diffusers and volutes. In order to reduce the “buzz-saw” noise of the air compressor, the back bend angles of the impeller outlet, the radius of the impeller outlets, the diffuser width, and the outlet area of the first stage volute are increased, so as to reduce the flow separation and restrain the disturbed flow within the volute tongue.

(3) After the structure optimization, the thermal performance of the air compressor maintains the same level, and the noise level is significantly reduced. The near-field sound pressure level is reduced by 5.3 dBA and 3.3 dBA at the first and second stage volutes respectively, and the near-field noise is reduced by 7.4 dBA at the inter-stage pipeline.

(4) There is an obvious noise attenuation effect on broadband noise and “buzz-saw” noise in the range of 0~4000 Hz by the application of the perforated muffler. The far-field sound pressure level reduces from 74.7 dBA to 68.9 dBA by 5.8 dBA under the rated condition.

The research in this paper realizes the key technologies of the whole chain from sound source identification, and induced mechanism exploration to noise control. The results provide the acoustic theoretical support and a practical basis for the forward development and upgrading of the air compressor for vehicle fuel cells.”

Comment 6: Finally, it is very difficult to understand what is actually new in the presented work and what is the knowledge gained. Very little literature is cited and compared to. There are many other works that should be referenced.

Response:

We would like to appreciate your suggestion. First, we have supplemented the references that you recommended. Second, we have reformulated the introduction section and highlighted the innovation of our work. The research results in most references show that the discrete single-tone noise at the blade passing frequency and its harmonics was more prominent and become the target of their research. However, the noise measurement of our research shows different results. The “buzz-saw” noise at shaft rotation frequency and its harmonics is far more prominent, as shown in figure 2. Our research provides a practical basis for the optimization of the “buzz-saw” noise of the vehicle fuel cell air compressors.

  • Sundstrom et al. (2014) "Assessment of the 3d flow in a centrifugal compressor using steady-state and unsteady flow solvers" compares two numerical approaches of the flow field simulation. The results show that both RANS and LES showed good agreement with the experimental measurements near the design condition, while using the RANS approach at off-design conditions lead to significant differences as compared to the experimental data. The investigated operating conditions of the compressor in our research are near the design points, RANS method is effective and low-cost for our research.
  • Sundstrom et al. (2018) "Acoustic signature of flow instabilities in radial compressors" employed LES to simulate the flow field and employed FW-H wave equation for the classification of acoustic sources. The results show that monopole and quadrupole were not as dominant. The acoustic source is classified at off-design conditions.
  • Freidhager et al. (2021) "Lighthill's analogy applied to an automotive turbocharger compressor" applied Lighthill's analogy to study the transient CFD simulations of turbocharger compressors. The method and the object of research in our paper are different.
  • Sharma et al. (2020) "Evaluation of modelling parameters for computing flow-induced noise in a small high-speed centrifugal compressor" quantified the impact of various turbulence formulations along with corresponding spatial and temporal resolutions on performance and acoustic predictions. The discrete single-tone noise at the blade passing frequency and its harmonics was as dominant, as shown in the figure.

Reviewer 2 Report

Page 2, Line 48: The author should mention how the operating condition and aerodynamic characteristics of this compressor differ from that are used in turbochargers using enough references

More elaborate on the whole system of vehicle fuel cell and the compressor in the system

provide a cross-section of compressors' components

do the impellers have splitters?

the real image of the test rig and the measuring devices are necessary. 

it is not clear what is idle, common, and rated conditions for the compressor. 

what are the rotational speeds and other characteristics?

please mention rotational speed, y+, mesh types, and quality, interface setting(moving and stationary part), how the time step is calculated,

the reported pressure ratio in Table 4, is for two stages.

the legend in Fig.7 is not readable

the reference plane for creating Fig.8 contours should be clarified

add dimensions for Fig 11

the first paragraph of page 10, (lines 266-272) needs some facts and figures from the simulation or some references

page 10, line 274, the structure of the com- 274 how the compressor is improved

Add a schematic for parameters in table 5

As I understood the optimization platform is developed in [15] but this  paper should address and introduced the parameters, their range, and the meta-model at least

adiabatic efficiency should be introduced

how the muffler is designed?

Author Response

Dear reviewer,

Thank you for your comments concerning our manuscript ID applsci-1921716 entitled " Study on Characteristics and Control of Aerodynamic Noise of a High-speed Centrifugal Air Compressor for Vehicle Fuel Cells". Those comments are very helpful for improving our paper. We have studied these comments carefully and also have tried our best to make corrections. The main corrections in the paper and the responses to the reviewer’s comments are as follows:

Comment 1: Page 2, Line 48: The author should mention how the operating condition and aerodynamic characteristics of this compressor differ from that are used in turbochargers using enough references.

Response:

Thank you for your comments and suggestions. We have supplemented a table to show the differences between the compressors used in turbochargers and the compressor we investigated. Please see the detail in the text as follows:

“Because its operating conditions and structural parameters are significantly different from those of the centrifugal air compressor used for turbochargers, which are shown in table 1, its aerodynamic noise characteristics and the improvement of the air compressor aimed at aerodynamic noise are worth investigating.

Table 1. Operating conditions and structural parameters of the compressors in references

References

Number of blades

Designed rotation speed/rev·min-1

Designed pressure ratio

Designed mass flow rate/kg·s-1

[1]

Main:13; Splitter:13

50000

4.0

2.8

[2]

8

4800

[3]

20

14000

5.32

[4]

32

2900

1.02

1.25

[5, 6]

Main:8; Splitter:8

22000

3.85

[9]

Main:7; Splitter:7

98529

0.211

Comment 2: More elaborate on the whole system of vehicle fuel cell and the compressor in the system. Provide a cross-section of compressors' components.

Response:

Thank you for your comments and suggestions. We have supplemented the elaboration on the whole system of the vehicle fuel cell and the cross-section of the compressor. Please see the detail in the text as follows:

“The research object of this paper is a two-stage centrifugal air compressor used in the air supply system for vehicle fuel cells. Figure 1 shows the schematic diagram of a typical hydrogen fuel cell system for vehicles. In order to increase the efficiency of the fuel cell, the air is compressed by a centrifugal compressor and then enters the fuel cell. Pressurized air reacts with hydrogen in the fuel cell to generate electric energy, which is supplied to the motor or battery.

Figure 1. Schematic diagram of a typical hydrogen fuel cell system for vehicles

The cross-section of the centrifugal compressor prototype is shown in figure 2.

Figure 2. Cross-section of the compressor”

Comment 3: Do the impellers have splitters?

Response:

Thank you for your kind question. The impellers of both two stages have no splitter. We have supplemented the statement in section 2.1.

Comment 4: The real image of the test rig and the measuring devices are necessary.

Response:

Thank you for your suggestion. We have supplemented the images in section 2.1. Please see the detail in the text as follows:

“The real images of the test rig and the sound level meter are shown in figure 4.

(a) Test rig

(b) Sound level meter

Figure 4. Real images of the test rig and the sound level meter”

Comment 5: It is not clear what is idle, common, and rated conditions for the compressor. What are the rotational speeds and other characteristics?

Response:

Thank you for your comments and questions. Different running states of vehicles require corresponding power provided by fuel cells. For higher fuel cell efficiency, the centrifugal compressor has corresponding speed changes to achieve different pressure ratios and mass flows rates. The detailed three operating conditions are shown in Table 3 in section 2. The three conditions are all near the design condition with different rotation speeds.

Comment 6: Please mention rotational speed, y+, mesh types, and quality, interface setting(moving and stationary part), how the time step is calculated.

Response:

Thank you for your suggestions. The rotational speed of the CFD model is corresponding to the operating condition shown in Table 3. The y+ of the mesh is set to 100 and the wall distance is 3.2e-4 m. The mesh in the impeller domain and diffuser domain is the structured hexahedron mesh. The mesh in the volute domain is the unstructured tetrahedral mesh. The minimum elements’ quality is above 0.2. The interface models between the moving and stationary parts are the frozen rotor for steady-state simulation and the transient rotor-stator for transient simulation. The boundary conditions for turbulence fractional intensity are 0.05 both at the inlet and the outlet. The Eddy length scales are 0.0386 m and 0.0394 m at the inlet and the outlet respectively. The time step is 1.852∙10-6 s, and the corresponding impeller rotates 1° at each time step under the rated speed. The detailed selection and verification of the time step refer to reference [9].

Comment 7: The reported pressure ratio in Table 4, is for two stages.

Response:

Thank you for your comments. We have clarified the definition of pressure ratio in section 3.1. Please see the detail in the text as follows:

“The pressure ratio (PR) of the compressor is for two stages and calculated as:

PR=Pout/Pin,

(1)

where the Pout is the pressure of the outlet of the 2ed volute and Pin is the pressure of the inlet of the 1st impeller.”

Comment 8: The legend in Fig.7 is not readable.

Response:

Thank you for your comments. We revised the legends in figure 7 for better reading. Please see the detail in the text as follows:

(a) First stage

(b) Second stage

Figure 9. Static entropy contours of 90% heights of impellers

(a) First stage

(b) Second stage

Figure 10. Static entropy contours of 50% heights of impellers”

Comment 9: The reference plane for creating Fig.8 contours should be clarified.

Response:

Thank you for your suggestions. The reference plane is at the half width of the diffuser and parallel to the bottom of the diffuser. We have supplemented the statement in section 3.1.

Comment 10: Add dimensions for Fig 11.

Response:

Thank you for your suggestions. We have added dimensions for figure 11. Please see the detail in the text as follows:

(a) First stage

(b) Second stage

Figure 16. Noise radiation directivity of centrifugal impellers”

Comment 11: The first paragraph of page 10, (lines 266-272) needs some facts and figures from the simulation or some references

Response:

Thank you for your suggestions. The flow separation results in a large number of vortex shedding and strong turbulence, which is regarded as the noise sound of the compressor. We have supplemented the citation of references[18,19], which shows the detailed relationship between flow separation and aerodynamic noise induction.

Comment 12: Page 10, line 274, the structure of the com- 274 how the compressor is improved. As I understood the optimization platform is developed in [15] but this paper should address and introduced the parameters, their range, and the meta-model at least.

Response:

We would like to appreciate your suggestion. The structure of the compressor is improved with the optimization platform. We have supplemented the introduction of the MOP model and the range of the parameters. Please see the detail in the text as follows:

“The aerodynamic optimization platform established in reference [21] is adopted for the parameter selection of the air compressor structure improvement in this paper. The optimization platform integrates geometric parameterization, surrogate model, sensitivity analysis, and multi-objective genetic algorithm. Among them, the sensitivity analysis mainly adopts the optimal prediction meta model (MOP), and the optimization process adopts the multi-objective genetic optimization algorithm based on the MOP surrogate model. Based on the optimal design variable subset of each target parameter, different surrogate models are established by polynomial method, moving least squares method, and Kriging method respectively. A set of competitive metamodels for each target parameter is formed. The evaluation is performed using a model-independent cross-validation method proposed by Most and Will, called the Coefficient of Prognosis. The value of the Coefficient of Prognosis is obtained based on the test sample point data of cross-validation. The prediction error is used to evaluate the prediction quality of the model. Both regression models and interpolation models are applicable. According to the size of the Coefficient of Prognosis of the meta-model, the optimal prediction meta-model of each target parameter is determined, that is, the MOP surrogate model.

Three target parameters are defined for the optimization object of this study, which are the total pressure loss coefficient, the average absolute airflow angle of the impeller outlet section, and the pressure range of the impeller outlet section. The ranges and changes of structural parameters before and after improvement are shown in Table 6.

Table 6. Ranges and changes in structural parameters before and after improvement

Structural parameters

Before improvement

Lower limit

Upper limit

After improvement

Back bend angle at the outlet of the first-stage impeller/°

52.0

50

54

52.5

Back bend angle at the outlet of the second-stage impeller/°

51.0

50

54

53.4

Outlet radius of the first-stage impeller/mm

33.1

26

40

37.1

Outlet radius of the second-stage impeller/mm

33.5

26

40

35.2

Width of the first-stage diffuser/mm

4.0

2

6

4.9

Width of the second-stage diffuser/mm

3.0

2

6

3.2

Outlet diameter of the first-stage volute/mm

36.8

30

44

41.6

Outlet diameter of the second-stage volute/mm

39.4

30

46

39.4

Comment 13: Add a schematic for parameters in table 5.

Response:

Thank you for your suggestions. We have supplemented a schematic of the optimized parameters. Please see the detail in the text as follows:

“The schematic of the optimized parameters is shown in Figure 17.

(a) Back bend angle

(b) Other optimized parameters

Figure 17. Schematic of the optimized parameters”

Comment 14: Adiabatic efficiency should be introduced.

Response:

Thank you for your suggestions. We have clarified the definition of pressure ratio in section 4.2. Please see the detail in the text as follows:

“The isentropic efficiency is calculated as:

(2)

where k is the specific heat ratio of air, Tout is the temperature of the outlet of the 2ed volute and Tin is the temperature of the inlet of the 1st impeller.”

Comment 15: How the muffler is designed?

Response: Thank you for your question. We have supplemented the introduction of the design process of the muffler. Please see the detail in the text as follows:

“The flow chart of the design for the muffler is shown in Figure 24. The general design principle is to maximize the resonant cavity within the confined compressor structure since larger resonant cavity has better noise reduction effect. Therefore, according to the structure of compressor, the geometric parameter is first selected to achieve a maximum cavity. Then, in order to ensure the structural strength and reliability, the thickness of the perforated wall is consistent with the thickness of the original structure. Afterwards, the noise reduction frequency is calculated and fine-tuned by adjusting the design parameters to meet the targeted noise reduction frequency. In the optimization process, the parameters of the proposed muffler are continuously adjusted, and the attenuation performance of the muffler is evaluated using 3-D simulation. When the attenuation frequency is consistent with the targeted attenuation frequency which is determined by the compressor operation characteristics, the attenuation pulsation is considered as maximum pressure pulsation attenuation and the selected parameters are determined. Since the muffler is designed according to the specific structure of the compressor, one of the design constraints is the compressor structure which limits the dimension and shape of the mufflers. Another design constraint is compressor operating condition which determines the specific frequency or specific frequency band.

Figure 24. Flow chart of the design method for the muffler”

Round 2

Reviewer 1 Report

The manuscript improved over the revision. However, the scientific impact remains questionable because of the lack of novelty presented. The manuscript represents more a work report than an academic article.

Author Response

Response to Reviewer 1 Comments

Point 1: The manuscript improved over the revision. However, the scientific impact remains questionable because of the lack of novelty presented. The manuscript represents more a work report than an academic article.

Response 1: We appreciate your comments. The logical framework of this paper is to improve the noise level of a centrifugal air compressor for vehicle fuel cells through technical means, so the layout of this paper is similar to the work report. We have revised the introduction section to highlight our novelty. We listed some of the innovations as follows:

  • Innovation of research objects: The operating conditions and structural parameters of the centrifugal air compressor for fuel cells are significantly different from those of the centrifugal air compressor used for turbochargers. The aerodynamic noise characteristics and the improvement of the air compressor aimed at aerodynamic noise are worth investigating.
  • Innovation of research methods: The application of independent component analysis in the identification of the sound source of centrifugal air compressors is innovative and practical.
  • Innovation of optimization target: The “buzz-saw” noise is tested as the most prominent component of aerodynamic noise in our research, which is inconsistent with the previous research conclusion. According to the difference between the characteristics of “buzz-saw” noise and discrete single-tone noise, we creatively applied a perforated muffler in the interstage pipe of the centrifugal air compressor.
  • Completeness of the research process: The research in this paper realizes the key technologies of the whole chain from the measurement of noise characteristics, sound source identification, and induced mechanism exploration to noise control. The research process can provide many references for other scholars or engineers. Subsequent scholars can also conduct in-depth research on one or more of these parts.

Thank you again for your comments. We will pay attention to your suggestion in our follow-up research reports.

Reviewer 2 Report

The revised manuscript improves significantly. However, it needs modification as well as an English language check.

1. In table 2, rows 3 and 4, 2nd is correct, not 2ed/ also in line 392

2. Why is the 2nd impeller's diameter bigger than the 1st one? Are you sure about this? After one compression stage, the compressed gas needs a smaller flow passage.

3. Please use [] for all dimensions that are mentioned in the tables

4. In Fig 11. on the legends, the word "contour 1" should be omitted.

5. Back bend angle (Fig. 17 a) is not a known term in rotary machines. Please use consistent terms in this regard. you can use: https://doi.org/10.3390/app9020291

 6. Add a nomenclature table for the manuscript. e.g., PR (pressure ratio) is not defined in the text. Is it total to total or not?

7. Show sensors on the test rig. For instance, indicate the flow meter, pressure taps locations, etc.

8. In Fig. 21 Idle condition is correct. 

Author Response

Response to Reviewer 2 Comments

Point 1: In table 2, rows 3 and 4, 2nd is correct, not 2ed/ also in line 392

Response 1: Thank you so much for your correction. We have revised all wrong spellings of 2nd.

Point 2: Why is the 2nd impeller's diameter bigger than the 1st one? Are you sure about this? After one compression stage, the compressed gas needs a smaller flow passage.

Response 2: Thank you for your question. We have checked the parameters of the compressor and revised table 2. Please see the detail in the text as follows:

Table 2. Main parameters of the compressor

Main parameters

First stage

Second stage

Diameter of impeller inlet [mm]

19.3

17.5

Diameter of impeller outlet [mm]

66.2

67

Width of impeller outlet [mm]

4

3

Number of the blades

15

15

As you said, after the compression of the first stage, the volume flow rate reduced from 0.111 m3·s-1 to 0.0814 m3·s-1. The flow passage of the second stage is smaller in actuality. The diameter of the second-stage impeller inlet is 0.0175 m, smaller than that of the first-stage impeller inlet. As a result, the flow area of the second-stage impeller inlet is smaller. Although the diameter of the second-stage impeller outlet is larger, the flow area of the second-stage impeller outlet is 831.9 mm2, smaller than that of the first-stage impeller, which is 631.5 mm2, because of the smaller width of the second-stage impeller outlet. The pressure ratio of the two stages of compressors is designed close. The larger outlet diameter of the second-stage impeller is the result of the comprehensive optimization design.

Point 3: Please use [] for all dimensions that are mentioned in the tables

Response 3: Thank you for your suggestion. We have revised all the tables in the paper.

Point 4: In Fig 11. on the legends, the word "contour 1" should be omitted.

Response 4: Thank you for your suggestion. We have omitted the word "contour 1" in Figure 11.

Point 5: Blade outlet angle (Fig. 17 a) is not a known term in rotary machines. Please use consistent terms in this regard. you can use: https://doi.org/10.3390/app9020291

Response 5: Thank you for your question. We have revised the blade outlet angle to the blade outlet angle.

Point 6: Add a nomenclature table for the manuscript. e.g., PR (pressure ratio) is not defined in the text. Is it total to total or not?

Response 6: Thank you for your question. We have supplemented a nomenclature. Please see the detail in the text as follows:

Nomenclature

CFD

computational fluid dynamics

BEM

boundary element method

ICA

independent component analysis

RANS

Reynold-averaged Naviers-Stokes

SST

shear stress transport

PR

pressure ratio

p

pressure

ηis

isentropic efficiency

k

specific heat ratio

T

temperature

The PR is the total to total. We have supplemented it in the definition of pressure.

Point 7: Show sensors on the test rig. For instance, indicate the flow meter, pressure taps locations, etc.

Response 7: Thank you for your suggestion. We have supplemented a legend in Figure 3 to better introduce the locations of sensors. Please see the detail in the text as follows:

Point 8: In Fig. 21 Idle condition is correct.

Response 8: Thank you for your suggestion. We have replaced the “idling condition” with “idle conditon”.
